# Stochastic resonance of rotating particles in turbulence

Ziqi Wang[1], Xander M. de Wit [1], Roberto Benzi [2,3] ✉, Chunlai Wu[1], Rudie P. J. Kunnen [1], Herman J. H. Clercx [1] & Federico Toschi [1,4] ✉

The chaotic dynamics of small-scale vorticity plays a key role in understanding and controlling turbulence, with direct implications for energy transfer, mixing, and coherent structure evolution. However, measuring or controlling its dynamics remains a major conceptual and experimental challenge due to its transient and chaotic nature. Here we use a combination of experiments, theory, and simulations to show that small magnetic particles of different densities, exploring flow regions of distinct vorticity statistics, can act as effective probes for measuring and forcing turbulence at its smallest scale. The interplay between the magnetic torque, from an externally controllable magnetic field, and hydrodynamic stresses, from small-scale turbulent vorticity, uncovers an extremely rich phenomenology. Notably, we present the first experimental and numerical observation of stochastic resonance for particles in turbulence, where turbulent fluctuations, remarkably acting as an effective noise, enhance the particle rotational response to external forcing. We identify a pronounced resonant peak in the particle rotational phase lag when the applied rotating magnetic field matches the characteristic intensity of small-scale turbulent vortices. Furthermore, we reveal a novel symmetry-breaking mechanism: an oscillating magnetic field with zero mean angular velocity can counterintuitively induce net particle rotation in zero-mean vorticity turbulence. Our findings pave the way to developing flexible techniques for manipulating particle dynamics in complex flows. The discovered mechanism enables a novel magnetic resonance-based approach to be developed for measuring turbulent vorticity. In this approach, particles act as probes, emitting a detectable magnetic field that can characterize turbulence even under conditions that are optically inaccessible.

Turbulent flows, characterized by chaotic multi-scale fluctuations, are central to a wide range of natural and industrial processes[1,2]. A key challenge in turbulence research is the direct measurement and control of the vorticity, which is primarily concentrated in small-scale vortex filaments[3,4]. Due to their transient nature and rapid chaotic evolution, capturing the rotational dynamics of these filaments remains a major conceptual and experimental challenge. Several methods have been developed to infer small-scale turbulence

[1]Fluids and Flows group and J.M. Burgers Center for Fluid Mechanics, Department of Applied Physics and Science Education, Eindhoven University of Technology, Eindhoven, Netherlands. [2]Sino-Europe Complex Science Center, School of Mathematics North University of China, Taiyuan, Shanxi, China. [3]Department of Physics and Istituto Nazionale di Fisica Nucleare, University of Rome Tor Vergata, Rome, Italy. [4]Consiglio Nazionale delle Ricerche - Istituto per le Applicazioni del Calcolo, Rome, Italy. ✉e-mail: roberto.benzi@gmail.com; f.toschi@tue.nl

characteristics, including particle tracking methods (e.g., tracer particles, deformable particles, and pattern-coated particles), as well as thermal and optical anemometry[5–7]. However, these approaches often suffer from inherent limitations such as insufficient spatial and temporal resolution, signal occlusion in optically dense environments, and difficulties in directly resolving vorticity at the smallest turbulence scales. As a result, achieving real-time, high-fidelity measurements of rotational motion in turbulent flows remains an ongoing challenge, particularly in complex and optically inaccessible flows.

A promising alternative is the use of magnetic particles, which respond to both stochastic vorticity fluctuations of turbulence (hydrodynamic torques[8]) and deterministic torques imposed by external magnetic fields. The dynamics of buoyant magnetic particles, in particular, are closely linked to the local vorticity. Particles with different densities exhibit distinct rotational behaviors as they explore different regions of the flow, leading to preferential concentration: light (heavy) particles tend to accumulate in high- (low-) vorticity regions[7,9,10]. These particles offer a unique dual role: they can serve as candidates for probing rotational fluctuations, while also enabling active flow manipulation through external forcing. This ability to modulate particle rotation and measure its response in real time provides a new pathway for investigating turbulence at small scales and suggests an elegant strategy for measuring turbulent vorticity. However, in order to unlock these novel directions, an accurate understanding of the complex interplay between hydrodynamic and magnetic torques on particle dynamics is required.

Previous studies have extensively explored magnetic particles in quiescent fluids under rotating magnetic fields, revealing rich phenomena such as self-assembly into chain-like structures[11,12], synchronization-selected structures[13], and more complex arrays[14], enhanced mixing[11,12,15], turbulence design[16–18] and phase locking, where particles follow the rotating magnetic field with a constant phase lag with respect to the magnetic field when the rotating magnetic angular velocity $\omega_H$ is below a threshold $\omega_{cr}$[19]. Investigations have also extended to self-propelling active magnetic particles (e.g., magnetotactic bacteria)[20,21], magnetic Janus particles[13,22], particles with different aspect ratios[23,24], colloidal spinners and active swimmers[25–27].

The competition and interplay between effects of turbulence and magnetic fields may lead to complex particle dynamics, yet to be understood, presenting a fundamental challenge in predicting and controlling particle behavior in turbulent environments.

Here, we explore this question by investigating the rotational dynamics of magnetic particles in turbulence under a rotating magnetic field. While deterministic synchronized and oscillation (back-and-forth) rotational regimes have been observed in zero-noise systems (e.g., compasses or rolling particles in quiescent fluids[28–30]), our work extends this understanding into the complex environment of fully developed turbulence. We systematically explore the complex interplay between deterministic external magnetic forcing and turbulent hydrodynamic torques, revealing a significantly richer phase space of particle rotational dynamics. We show that the effect of turbulence (vorticity) acts as an effective noise (Supplementary Information) and reveal the first signature of the emergence of stochastic resonance (SR)—a counterintuitive phenomenon where internal fluctuations non-linearly cooperate with external forcing to amplify the system response[31–39]—of particles in turbulence, which can be used as the mechanism of magnetic particles acting as effective vorticity probes for a novel type of turbulence "microscope". While the concept of SR, initially observed in bistable systems subjected to noise and weak periodic signals, has since been extended to a wide range of physical and biological systems, as well as signal processing and computer science[40–45], its role in turbulence-driven rotational dynamics remains mostly unexplored.

To identify potential SR signatures, we set out to leverage turbulent vorticity fluctuations as a source of noise and examine the response of the magnetic particle dynamics to the rotating magnetic field, by means of experiments, simulations, and theoretical modeling. Our experimental platform (Fig. 1a and e) involves a dilute collection of approximately spherical light magnetic particles (produced by Styrofoam core coated with magnetic paint, Fig. 1b, with a mean density of 20% with respect to water and with a mean diameter of 0.7 mm, see "Methods" for details) which are immersed in a Von Kármán-type turbulent flow (producing homogeneous and isotropic turbulence (HIT) in the center region of the cell[46–49]) in an octagonal water-filled chamber. Magnetic particles are designed to be light and of a size comparable to the Kolmogorov scale, allowing them to preferentially enter and interact with vortex filaments. The flow is generated by two counter-rotating bladed disks and the chamber is placed in a system of Helmholtz coils, which is programmed to generate a uniform magnetic field in a restricted measurement volume, $\mathbf{H} = H\mathbf{h}$, rotating in the $xoy$-plane with angular velocity vector $\boldsymbol{\omega}_H = \omega_H \hat{\mathbf{z}}$ where $\hat{\mathbf{z}}$ is the unit vector along the $z$-axis ("Methods" and Supplementary Information). As the induced magnetic dipole, $\mathbf{m}$, of a particle tends to align with the magnetic field direction, $\mathbf{h}$, the particle experiences a magnetic torque balanced by the rotational drag and spins around an axis in the particle body frame parallel to $\hat{\mathbf{z}}$ along its trajectory (as shown by typical stroboscopic time-lapse trajectories in Fig. 1c, d). The rotational motion is measured by tracking particle surface patterns ("Methods").

## Results
### Theoretical representation

The particle rotation is characterized by a preferred magnetization direction, $\mathbf{n}$ (Fig. 2a), whose trajectory can be tracked in turbulence under the influence of a rotating magnetic field. By balancing the magnetic torque and drag torque induced by turbulence[19,50] ("Methods" and Supplementary Information), the equations governing the particle rotational motion can be obtained, in the overdamped limit, as

$$\frac{d\mathbf{n}}{dt} = \omega_a (\mathbf{n} \cdot \mathbf{h})(\mathbf{h} - (\mathbf{n} \cdot \mathbf{h})\mathbf{n}) + \boldsymbol{\omega}_f \times \mathbf{n}, \tag{1a}$$

$$\frac{d\left(\xi_r \boldsymbol{\omega}_{p,\parallel}\right)}{dt} = -\xi_r \left(\boldsymbol{\omega}_{p,\parallel} - (\boldsymbol{\omega}_f \cdot \mathbf{n})\mathbf{n}\right), \tag{1b}$$

where $\omega_a = \mu \mathcal{V}_p H^2 (\chi_\parallel - \chi_\perp)/\xi_r$ is the magnetic field strength normalized by the rotational drag coefficient $\xi_r$, with $\mu$, $\mathcal{V}_p$, and $\chi_{\parallel,\perp}$ the particle permeability, volume, and anisotropic magnetic susceptibilities, respectively[19,50,51] ("Methods"). The term $\boldsymbol{\omega}_{p,\parallel}$ represents the particle spinning rate. The perpendicular rotational component, $\boldsymbol{\omega}_{p,\perp}$, can be obtained by $\boldsymbol{\omega}_{p,\perp} = \mathbf{n} \times \dot{\mathbf{n}}$ which represents the particle tumbling rate. The turbulent vorticity experienced by the particle is $\boldsymbol{\omega}_f$, which is defined as $\boldsymbol{\omega}_f = \frac{1}{2}[\nabla \times \mathbf{u}_f]_O$ with the subscript O implying evaluation at the center of mass of the particle. The total particle angular velocity is thus $\boldsymbol{\omega}_p = \boldsymbol{\omega}_{p,\parallel} + \boldsymbol{\omega}_{p,\perp}$.

In a noiseless system, i.e., $\boldsymbol{\omega}_f = \mathbf{0}$, the critical frequency is $\omega_{cr} = \omega_a/2$ (details in Supplementary Information). For $\omega_H \leq \omega_{cr}$, the particle is synchronized with the rotating magnetic field ("phase-locked"), i.e., $\omega_{p,z} = \omega_H$. Otherwise, when $\omega_H > \omega_{cr}$, the particle undergoes back-and-forth motion ("back-and-forth") when the maximum magnetic torque cannot balance the response drag torque: the phase lag angle $\beta$ between $\mathbf{n}_{xoy}$ (the projection of $\mathbf{n}$ on $xoy$-plane) and $\mathbf{h}$ changes periodically over time[20]. These two regimes can be observed experimentally (Fig. 1f–i). In the "phase-locked" regime (Supplementary Movie 1), the particle rotates synchronously with the magnetic field, as evidenced by the normalized angular velocity, $\omega_{p,z}/\omega_H \approx 1$, in Fig. 1g. In the "back-and-forth" regime (Supplementary Movie 2), the particle oscillates periodically (Fig. 1i).

As illustrated in the schematic diagram of Fig. 2a, the dynamics of the particle preferred magnetization direction, $\mathbf{n}$, is fundamentally

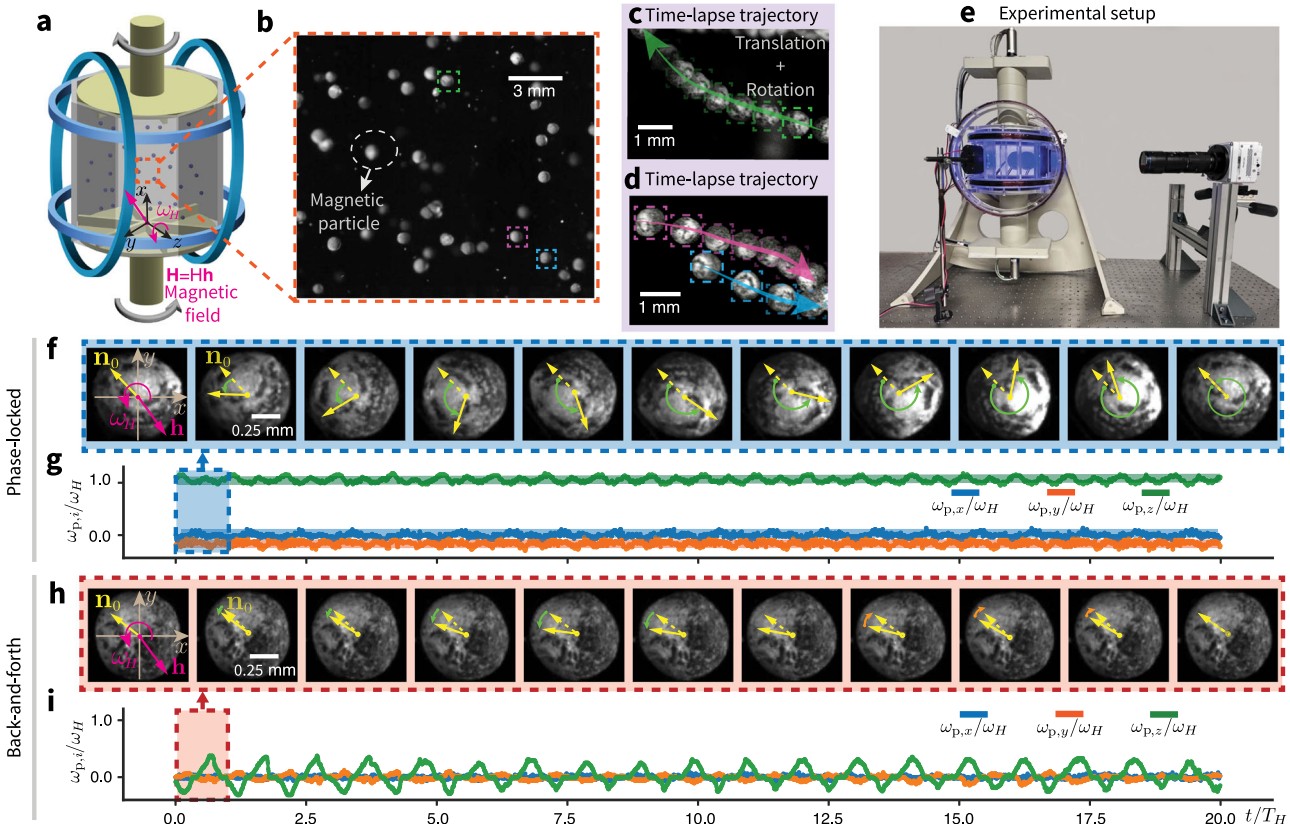

**Fig. 1 | Rotational dynamics of magnetic particles in a rotating magnetic field: "phase-locked" versus "back-and-forth" regimes. a** Magnetic particles in turbulence under a rotating magnetic field: The experimental setup consists of a turbulence generator, a magnetic field generator, and magnetic particles. A Von Kármán-type turbulent flow is generated in an octagonal water-filled container with an internal diameter of $2R = 150$ mm and a height of 220 mm, driven by two counter-rotating bladed disks. A uniform rotating magnetic field $\mathbf{H} = H/\mathbf{h}$, with angular frequency $\omega_H$, is generated using a system of two pairs of perpendicular Helmholtz coils. **b** Magnetic particles consist of a Styrofoam core coated with magnetic paint. Surface patterns enable tracking rotational motion, as illustrated in

the stroboscopic time-lapse trajectories (**c**, **d**). **e** Photograph of the setup. **f–i** In a quiescent fluid, particles exhibit two rotational regimes: "phase-locked" (low $\omega_H$) and "back-and-forth" (high $\omega_H$). **f** (Supplementary Movie 1) and **h** (Supplementary Movie 2) show rotational trajectories within one rotation period, viewed in a plane perpendicular to the rotation plane of the magnetic field. The particle initial preferred magnetization direction $\mathbf{n}_0$ (dashed arrows) and instantaneous orientation $\mathbf{n}$ (solid arrows) are marked over time. During rotation within a quiescent fluid, $\mathbf{n}$ and $\mathbf{h}$ remain within the same plane[50]. The corresponding normalized particle angular velocity, $\omega_{\mathrm{p},i}/\omega_H$ ($i = x, y, z$), is shown in (**g** and **i**). Detailed information about the experiments and simulations can be found in "Methods".

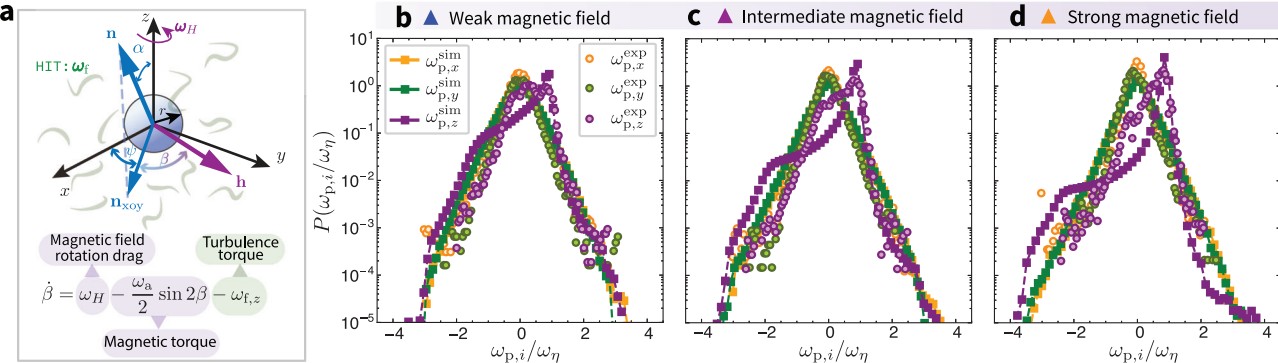

**Fig. 2 | Experimental validation of the theoretical model for particle rotational dynamics with turbulence. a** Schematic of a magnetic particle in turbulence under a rotating magnetic field. The particle with preferred magnetization direction $\mathbf{n}$ is subjected to a magnetic field $\mathbf{h}$ in the $xoy$-plane, rotating around the $z$-axis with frequency $\omega_H$, and experiences turbulent vorticity $\boldsymbol{\omega}_f$ in homogeneous isotropic turbulence. The phase lag $\beta$ (defined as the angle between the projection of $\mathbf{n}$, i.e., $\mathbf{n}_{\mathrm{xoy}}$, and $\mathbf{h}$) and the rotation angle $\psi$ (angle of $\mathbf{n}_{\mathrm{xoy}}$ relative to the $x$-axis) describe the particle dynamics. The dynamical evolution of $\beta$ is governed by the interplay of

the magnetic field rotation drag, magnetic torque, and turbulence torque. **b–d** Comparison of the probability density distribution function (PDF) of particle angular velocity in experimental (circles) and numerical (squares) studies. The experimental results (circles) are shown for varying magnetic field strengths with fixed turbulence intensity. **b** weak (1.2 mT, Supplementary Movie 3), **c** intermediate (1.4 mT, Supplementary Movie 4), and **d** strong (1.6 mT, Supplementary Movie 5), with constant rotational frequency of the magnetic field (parameter settings can be found in the "Methods" and Supplementary Information).

characterized by the phase lag angle, $\beta$, with respect to $\mathbf{h}$ on the $xoy$-plane. The governing equations, Eqs. (1), can be simplified and explicitly written as (derivations in "Methods")

$$\dot{\beta} = \omega_H - \frac{\omega_a}{2}\sin 2\beta - \omega_{f,z}. \tag{2}$$

Geometrically, we have $\beta = \omega_H t - \psi$. The phase lag angle $\beta$ represents the particle rotational displacement in the co-rotating frame of the magnetic field, while $\psi$ is the rotation angle of the projection $\mathbf{n}$ on the $xoy$-plane with respect to the $x$-axis, which describes the particle rotational displacement in the laboratory frame. The turbulent vorticity signal, $\boldsymbol{\omega}_f$, represents the vorticity experienced by the particle center along its trajectory. This simplified representation, Eq. (2), separates different contributions and clearly shows that the rotational motion of the particle results from the interplay between the magnetic field rotational drag, the magnetic torque, and the turbulence drag torque (Fig. 2a).

## Switching on turbulence

Now we wonder how this picture will change when turbulence is taken into consideration. The turbulence induces fluctuations on the small particles of a typical magnitude $\omega_\eta = 1/\tau_\eta$, the Kolmogorov characteristic frequency. Experimentally, we turn on the turbulence and vary the intensity of the magnetic field. By statistically analyzing the probability density function (PDF) of the particle angular velocity $\boldsymbol{\omega}_p$, a dominant peak is observed in the $z$-direction at approximately $\omega_{p,z} \approx \omega_H$ (Fig. 2b–d, purple circles), while the angular velocities in the $x$ and $y$ directions are similar to each other but significantly different from that in the $z$-direction (Fig. 2b–d, yellow and green circles).

To understand the influence of turbulence on the particle dynamics further, we carry out a systematic numerical investigation. To model the motion of light particles in HIT, we first perform direct numerical simulations using a pseudo-spectral method[10,52,53] to obtain $\boldsymbol{\omega}_f$ (see "Methods" for details). The resulting vorticity data is then incorporated into the theoretical model, Eqs. (1), which is integrated using a third-order Runge–Kutta method.

The simulated particle rotation is validated against the experimental results (Fig. 2b–d and Supplementary Movies 3–5, respectively). Notably, the theoretical model (squares) accurately captures the essential statistical physics (peak positions) of the particle stochastic rotational dynamics in the experiments (circles). The slight discrepancies in the PDF shapes arise from polydispersity in particle shape and magnetic properties in experiments which influences how individual particles respond to an external magnetic field, whereas the simulations assume idealized spherical particles with the same magnetic characteristics for each particle ("Methods" and Supplementary Information). Further comparisons between simulation and experimental results all demonstrate good qualitative agreement (Supplementary Information, Supplementary Movies 9 and 10).

Additionally, to gain deeper insight into the stochastic nature of particle rotation in turbulence, we analyze the waiting time, $\tau$—the duration between successive transitions of $\beta$ between its metastable states—to characterize the nature of turbulence-induced noise in the absence of external forcing (i.e., $\omega_H = 0$). The Poisson-like distribution of $\tau$ confirms that, within the investigated parameter space, stochastic transitions induced by turbulent (vorticity) fluctuations can be effectively modeled using white noise with an equivalent intensity (Supplementary Information).

## Three distinct regimes of particle dynamics

With the theoretical model capturing the core dynamics of this complex system, and with simulations allowing exploration of a broader parameter space and finer details of particle rotation, we systematically investigate how turbulence-induced vorticity fluctuations and the deterministic magnetic forcing shape particle rotational dynamics.

Beyond the previously identified "phase-locked" and "back-and-forth" regimes, we uncover a third regime: the turbulence-dominated regime, observed at small $\omega_a$ w.r.t. $\omega_\eta$, when the effect from the magnetic field is negligible. Here, the particle rotation is primarily driven by turbulence, and its normalized angular velocity, $\omega_p/\omega_H$, fluctuates randomly around zero with no net angular drift (Fig. 3b), as evidenced by the random orientation distribution of the particle orientation, $\mathbf{n}$ (Fig. 3d and Supplementary Movie 6). Consequently, the phase lag $\beta$ and its derivative are also randomized (Fig. 3c). In the "phase-locked" regime ($\omega_a \gg \omega_\eta$, $\omega_H < \omega_{cr}$), the particle phase lag, $\beta$, remains constant, meaning $\dot{\beta} = 0$ (Fig. 3g). The particle angular velocity along the magnetic field rotation axis ($z$-axis) is locked to $\omega_H$ (Fig. 3f, green line). The particle orientation is strongly confined, exhibiting 2D behavior (Fig. 3h and Supplementary Movie 7). In the "back-and-forth" regime ($\omega_a \gg \omega_\eta$, $\omega_H > \omega_{cr}$), the particle undergoes periodic acceleration and deceleration (Fig. 3j, k, inset), forming distinct looping patterns in its orientation trajectory (Fig. 3l and Supplementary Movie 8). These regimes are summarized in a phase diagram, characterized by the normalized variance of the particle angular velocity in the $z$-direction, $\langle \omega_{p,z}^2 - \langle \omega_{p,z} \rangle^2 \rangle / \omega_\eta^2$, where $\langle \cdot \rangle$ represents a time average. The noiseless critical frequency $\omega_{cr} = \omega_a/2$ is marked by a black dot-dashed line. Unlike in a quiescent fluid, the phase-locked regime (Fig. 3m, purple region) is bounded by the blue line, predicted by the scaling argument based on the simplified governing equation of Eq. (2): $\omega_a = 2\omega_H + \omega_\eta$ (detailed reasoning can be found in "Methods"). This shift in the subcritical boundary, relative to the noiseless case ($\omega_a = 2\omega_H$), accounts for the influence of turbulence fluctuations, which have a standard deviation on the order of $\omega_\eta$.

## Stochastic resonance

To determine whether SR occurs, we focus on the $\beta$ and its behavior across different parameter settings. We summarize the time-averaged value of $\langle \dot{\beta} \rangle / \omega_\eta$ in one phase diagram (Fig. 4a), with $\omega_\eta/\omega_a$ (vertical axis) quantifying the relative turbulent fluctuation intensity (noise) to the applied magnetic field strength, and $\omega_H/\omega_a$ (horizontal axis) distinguishing noiseless subcritical, critical (i.e., $\omega_H/\omega_a = 1/2$, black dot-dashed line), and supercritical states.

For a fixed $\omega_H/\omega_a$ in the subcritical regime ($\omega_H/\omega_a < 1/2$), the particle rotation in the co-rotating frame $\langle \dot{\beta} \rangle / \omega_\eta$ exhibits a non-monotonic dependence on noise intensity (Fig. 4b), with a pronounced resonant peak at $\omega_\eta/\omega_a \approx 1$, indicating that resonance arises at the transition from the "phase-locked" regime to the "turbulent-dynamics" regime. This is a clear suggestion that the observed resonance behavior results from the cooperation between turbulent fluctuations and magnetic forcing. SR is also clearly observed experimentally, as shown in Fig. 4c: diamonds denote the experimental results at $\omega_H/\omega_a = 0.12 \pm 0.01$ (corresponding to the points marked in the phase diagram in Fig. 4a), while circles represent the simulation results at a comparable value of $\omega_H/\omega_a$ for direct comparison. The experimental and simulation results agree reasonably well; at the resonance peak, the experimental curve appears flatter which can be attributed to effects of polydispersity of the particles used in experiments. The resonance mechanism can be interpreted via an effective potential well framework (Fig. 4d), where the magnetic field induces a cosine-type potential (i.e., $-\frac{\omega_a}{4}\cos 2\beta$) modulated by a linear term (i.e., $-\omega_H\beta$), while turbulence acts as a stochastic noise.

For $\omega_H/\omega_a < 1/2$ (Fig. 4e), the phase lag angle is initially trapped in the potential minimum at weak noise levels (small $\omega_\eta/\omega_a$), exhibiting only mild fluctuations locally. As the turbulence-induced fluctuations become comparable to the magnetic field strength ($\omega_\eta/\omega_a \approx 1$), preferential unidirectional escape occurs. This clearly identifies the observed resonant behavior as an example of SR: a cooperative interplay between stochastic forcing and the external deterministic forcing. This phenomenon can be thought of as the particle "surfing" the rotating magnetic field, aided by the turbulent vorticity. At even higher noise levels, randomization overrides the cooperative effect, causing bidirectional escape and a decline in $\langle \dot{\beta} \rangle$.

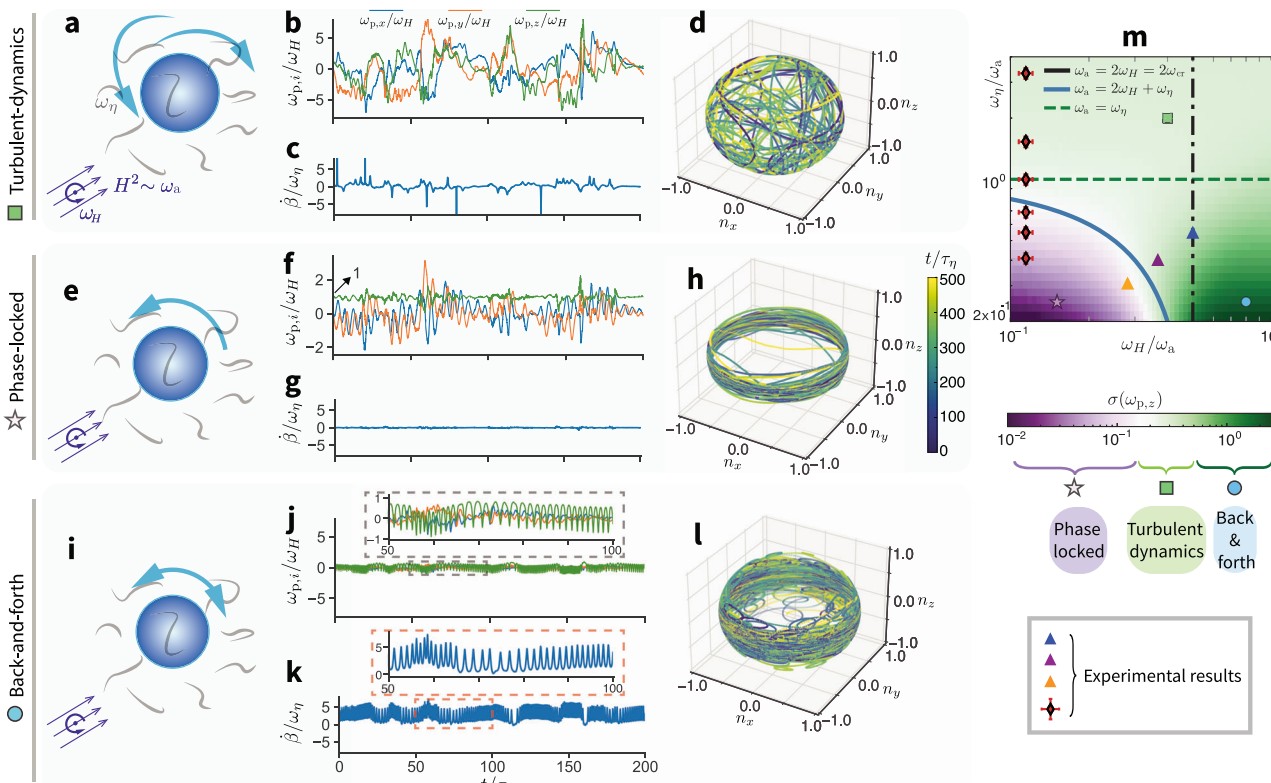

**Fig. 3 | Three distinct regimes of magnetic particle rotation dynamics in turbulence under a rotating magnetic field. a** Turbulence-dominated regime ($\omega_a \ll \omega_\eta$). **b** The normalized particle angular velocity, $\omega_p/\omega_H$, exhibits turbulent fluctuations with a zero-average and the phase lag derivative, $\dot{\beta}$ is randomized (**c**). **d** The evolution of the tip of preferred magnetization direction of the particle, **n**, is visualized in space, with color indicating time progression (Supplementary Movie 6). The random distribution of orientations confirms the turbulent-dominated nature. **e** "Phase-locked" regime ($\omega_a \gg \omega_\eta$, $\omega_H < \omega_{cr}$): The angular velocity of the particle along the magnetic field rotation axis (z-axis) is locked to $\omega_H$ (**f**, green line), with zero net drift for other components (**f**, blue and orange lines). The phase lag derivative $\dot{\beta}$ is effectively zero (**g**) and the particle orientation follows a 2D trajectory (**h**, Supplementary Movie 7). **i** "Back-and-forth" regime ($\omega_a \gg \omega_\eta$, $\omega_H > \omega_{cr}$): The angular velocity of the particle along the z-axis exhibits periodic

acceleration and deceleration oscillations (**j**, inset), and $\dot{\beta}$ varies similarly (**k**, inset). The orientation vector oscillates periodically (back-and-forth looping in **l**, Supplementary Movie 8). **m**, Phase diagram of particle rotation regime, colored by the normalized variance $\sigma(\omega_{p,z})$ of the particle angular velocity along the z-direction, $\sigma(\omega_{p,z}) = \langle \omega_{p,z}^2 - \langle \omega_{p,z} \rangle^2 \rangle / \omega_\eta^2$. Black dot-dashed line: the critical frequency, $\omega_{cr} = \omega_a/2$. Green dashed line: the condition where the turbulence fluctuation intensity is balanced by the magnetic strength, i.e., $\omega_a = \omega_\eta$. The phase-locked regime is bounded with the blue thick line, predicted by the scaling analysis, i.e., $\omega_a = 2\omega_H + \omega_\eta$. Symbols (square, star, and circle): representative cases from the three distinct regimes shown in earlier panels. Triangle markers: experimental settings from Fig. 2b–d. The diamond symbols denote the experimental results at $\omega_H/\omega_a = 0.12 \pm 0.01$, providing evidence of stochastic resonance, which will be discussed further in Fig. 4a and c.

At the critical threshold $\omega_H/\omega_a = 1/2$, the potential well vanishes, rendering $\beta$ critically sensitive to any perturbations (Fig. 4g). In the supercritical range ($\omega_H/\omega_a \gg 1/2$), the potential well inherently favors persistent phase slipping, regardless of noise intensity (Fig. 4h), such that the SR mechanism vanishes in this regime.

In strongly subcritical regime ($\omega_H/\omega_a \ll 1/2$), the resonance peak is purely set by the turbulent noise, making it sensitive to the preferential sampling effect[7,9,10] (see Supplementary Information for neutral and heavy particles results). Here we introduce the relaxation time $\tau^* = 1 / \left( \omega_a \sqrt{1 - 4\left(\frac{\omega_H}{\omega_a}\right)^2} \right)$. Physically, $\tau^*$ represents the characteristic time it takes for a perturbed particle rotational motion to return to its phase-locked state, i.e., $\dot{\beta} = 0$, after a small disturbance (detailed derivations are in Supplementary Information). Close to the critical ratio $\omega_H/\omega_a = 1/2$, the significantly increasing particle relaxation time $\tau^*$ shifts the resonance position, making it the dominant factor. Therefore, a longer $\tau^*$ allows particles to escape the potential well even at lower noise levels.

### Symmetry breaking mechanism

A striking consequence of SR is the symmetry-breaking effect, manifesting as a "zero plus zero is greater than zero" phenomenon. This

leverages the fact that in the weak noise regime ($\omega_\eta < \omega_a$), the response of the system to the noise is non-linear (Fig. 4b, purple-shaded part).

To reveal this effect, instead of applying a constant-frequency rotating magnetic field ($\omega_H$), we now impose a time-dependent field rotation with a zero-average, $\omega_H(t)$ (Fig. 5a). Each oscillation cycle consists of a forward and a reverse rotation phase, characterized by angular velocities $\omega_H^+$ and $\omega_H^-$ with corresponding durations $T_H^+$ and $T_H^-$. These parameters are scaled as $\omega_H^+/\omega_H^- = T_H^-/T_H^+ = \gamma$, ensuring that the net angular velocity over a full cycle remains zero. Two distinct response regimes emerge as $\omega_\eta/\omega_a$ increases for a fixed $\gamma$ (Fig. 5b), highlighting the ability of this oscillation framework to probe the nonlinearity of the system in response to noise. Specifically, the response $\langle \dot{\beta} \rangle/\omega_\eta$ decreases monotonically in the nonlinear regime (weak noise limit), while it remains nearly constant and close to zero in the linear regime (strong noise limit, characterized by response collapse, see also Fig. 4b, inset). This behavior aligns well with a superposition model, which predicts the response as

$$\langle \dot{\beta} \rangle = \frac{\langle \dot{\beta}(\omega_H^+) \rangle T_H^+ - \langle \dot{\beta}(\omega_H^-) \rangle T_H^-}{T_H}, \quad (3)$$

with $T_H = T_H^+ + T_H^-$, where $\langle \dot{\beta}(\omega_H^+) \rangle$ and $\langle \dot{\beta}(\omega_H^-) \rangle$ are interpolated from the numerical results in Fig. 4b. The predicted trends (Fig. 5b, dashed

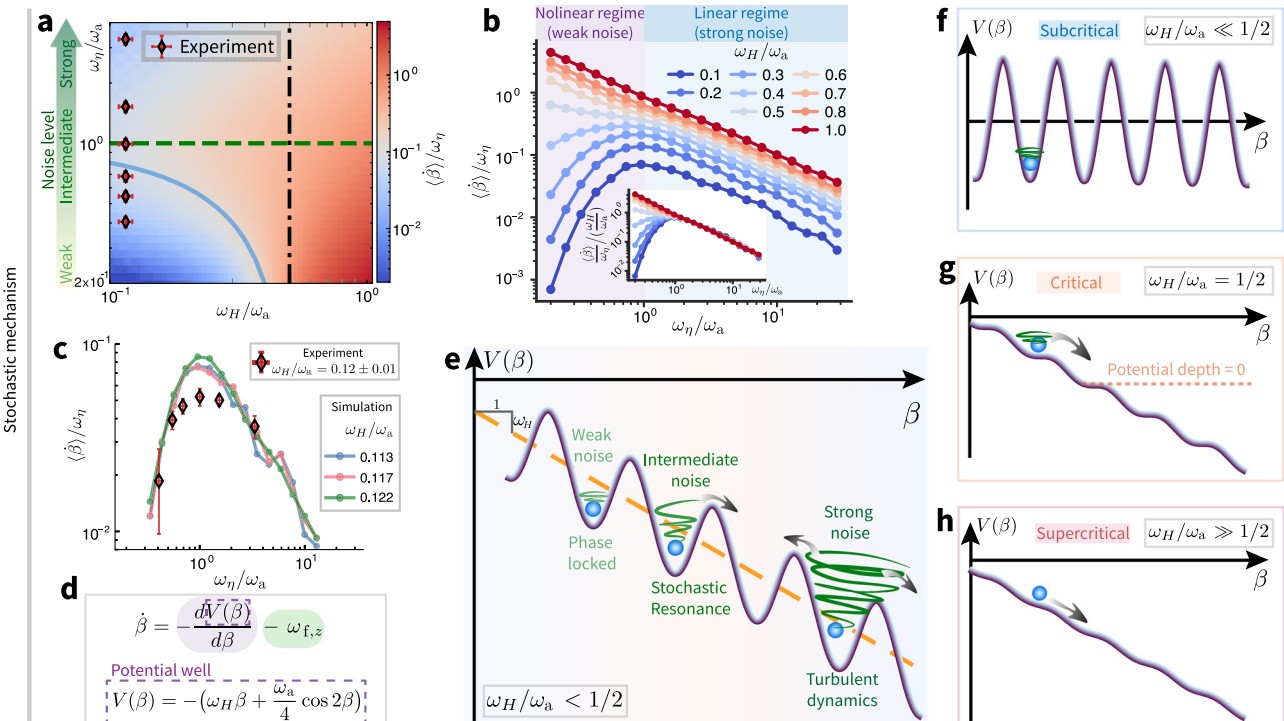

**Fig. 4 | Stochastic resonance. a** Phase diagram of stochastic resonance. The color represents the normalized time-averaged phase lag derivative, $\langle\dot{\beta}\rangle/\omega_\eta$. The vertical axis, $\omega_\eta/\omega_a$, quantifies the relative intensity of turbulent fluctuations (noise) to the magnetic field strength. The horizontal axis, $\omega_H/\omega_a$, distinguishes subcritical, critical ($\omega_H/\omega_a = 1/2$, marked by the black dot-dashed line) and supercritical regimes. For a fixed $\omega_H/\omega_a$ in the subcritical (supercritical) range, $\langle\dot{\beta}\rangle/\omega_\eta$ exhibits a non-monotonic (monotonic) dependence on noise. A pronounced resonant peak appears at $\omega_\eta/\omega_a \approx 1$ when $\omega_H/\omega_a < 1/2$ (**b**), with a perfect collapse in the linear regime when normalized by $\omega_H/\omega_a$ (**b**, inset). The diamond symbols in (**a**) denote the parameter settings for experiments shown in (**c**). **c** Experimental evidence of stochastic resonance (diamonds) at $\omega_H/\omega_a = 0.12 \pm 0.01$. For comparison, simulation results at a similar value of $\omega_H/\omega_a$ (circles) are also shown. The error bars

correspond to the standard deviation calculated from 20 independent measurements at the same parameter value. **d** Effective potential well framework: the magnetic field contributes a cosine potential modulated by a linear term, and turbulence acts as a stochastic noise, see Eq. (2). When $\omega_H/\omega_a < 1/2$ (**e**), for weak noise, the phase lag remains near the local minimum of the potential well. As noise increases, the particle phase lag exhibits unidirectional escape, signaling stochastic resonance. At even higher noise levels, escape becomes bidirectional, leading to a subsequent decline in $\langle\dot{\beta}\rangle$. For $\omega_H/\omega_a \ll 1/2$, the potential well is sufficiently deep to strongly localize the particle phase lag (**f**). At the critical threshold $\omega_H/\omega_a = 1/2$, the potential well vanishes, making the phase lag sensitive to small perturbations (**g**). In the supercritical range ($\omega_H/\omega_a \gg 1/2$), the potential well inherently favors persistent phase slipping, regardless of noise intensity (**h**).

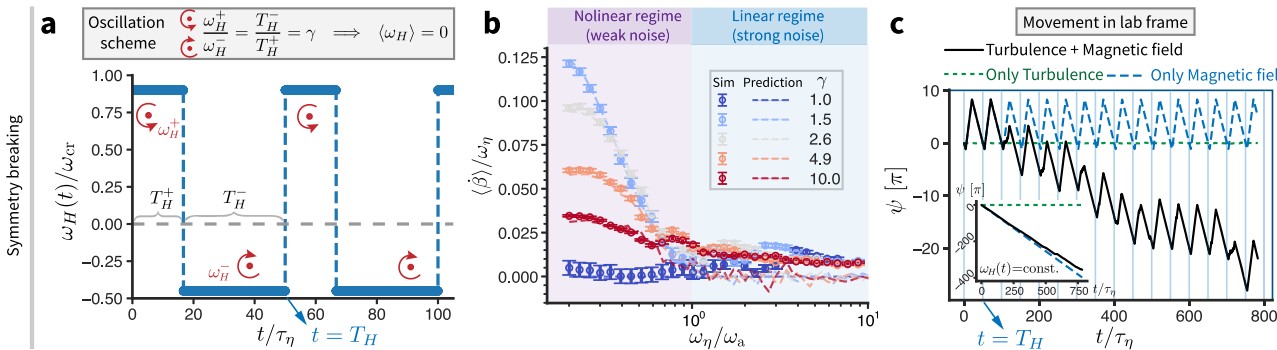

**Fig. 5 | Symmetry breaking. a** The applied magnetic field oscillates with zero-average angular velocity, $\omega_H(t)$. Without loss of generality, we set $\omega_H^+/\omega_H^- = T_H^-/T_H^+ = \gamma \geq 1$ and $\omega_H^+ = 0.9\omega_{cr}$. **b** The normalized phase lag derivative, $\langle\dot{\beta}\rangle/\omega_\eta$, shows distinct regimes as noise intensity $\omega_\eta/\omega_a$ increases for fixed $\gamma$. In the nonlinear regime (low noise), $\langle\dot{\beta}\rangle/\omega_\eta$ decreases monotonically, while in the linear regime (strong noise, where system responses collapse, see Fig. 4b, inset), it remains nearly unchanged and close to zero. When $\gamma = 1$, the particle rotation motions cancel each other out in $T_H^-$ and $T_H^+$, resulting in a zero response. The observed deviation from a strictly zero value in the simulations is attributable to the finite periods of the applied oscillation cycles of $\omega_H(t)$. Theoretically, with an infinite simulation time, the value would converge to zero. The error bars represent the standard deviation of the results obtained by sampling over one full period of $\omega_H(t)$

at the final stage of the simulation. The predicted trends by the superposition model of Eq. (3) (**b**, dashed lines) exhibit excellent agreement with the numerical data (**b**, symbols). **c** Particle motion in laboratory frame. With turbulence alone (green dotted line), the particle shows no net angular movement in terms of $\psi$; with only the magnetic field (blue dashed line), the motion is periodic with zero net angular displacement. When both effects act together, a net angular movement emerges (black solid line), revealing symmetry-breaking mechanism. This is different from the case of constant $\omega_H$ (inset), where the particle follows $\psi = \omega_H t$ (black solid line) when it is purely driven by a magnetic field. Parameter settings are $T_H = 50\tau_\eta \approx T_L$ with $T_L$ the integral time scale of turbulence, $\omega_\eta/\omega_a = 0.25$, and $\gamma = 2$. Inset: $\omega_\eta/\omega_a = 0.25$ and $\omega_H/\omega_a = 0.4$.

lines) show excellent agreement with the frequency-modulated field results (Fig. 5b, symbols).

We emphasize that when combining turbulence and the frequency-modulated magnetic field in this way, this leads to particle motion directly in the lab frame. To evidence this, we consider the lab-frame angular particle motion $\psi(t)$ (Fig. 5c). With turbulence alone, the particle exhibits no net angular movement (green dotted line). Under a purely frequency-oscillatory magnetic field, the particle undergoes periodic motion with zero net angular displacement (blue dashed line). However, when both effects act concurrently, a net, non-zero mean angular drift remarkably emerges (black solid line), as the turbulent vorticity aids the particle in "surfing" the magnetic field, which reveals an unexpected symmetry-breaking mechanism in turbulent flows. This stands in contrast to the case of a constant $\omega_H$ (Fig. 5c, inset), where a purely magnetic field-driven particle, instead of showing zero net angular movement as in the oscillating $\omega_H$ scheme, follows $\psi = \omega_H t$ (black solid line).

## Discussion

We have revealed the first experimental and numerical evidence of SR for magnetic particles suspended in turbulence, demonstrating how the interplay between turbulent fluctuations and external forcing induces coherent rotational motion of magnetic particles. Three distinct regimes: "phase-locked", "back-and-forth", and "turbulent-dynamics" are identified, with SR occurring at the transition between "phase-locked" and "turbulent-dynamics" regimes. Remarkably, even when both turbulence and the magnetic field have zero mean angular velocity, their collective effect is capable of generating a nonzero particle rotational response, as the turbulent vorticity helps the particle to "surf" the rotating magnetic field.

Different particle properties influence their response, and when combined with the SR mechanism, they enable a novel magnetic resonance-based approach to measure turbulent vorticity. This method leverages activated magnetic particles as probes, effectively functioning as a turbulent vorticity "microscope" that operates in various conditions. In the present work, we validate this concept using optical measurements of particle rotation. However, in principle, once the angular response is well-characterized, the same protocol could be applied even in optically inaccessible environments by measuring the magnetic fields emitted by the spinning magnetic particle directly. This concept shares similar principles with Magnetic Particle Spectroscopy, a technique that uses the dynamic magnetic responses of magnetic particles to remotely sense properties of their surrounding environment[54]. In such scenarios, the activated magnetic particles would function as remotely readable vorticity probes. The protocol is straightforward: by dispersing magnetic particles into turbulence and varying the magnetic frequency $\omega_H$ and intensity $\omega_a$ while maintaining a constant ratio $\omega_H/\omega_a$, one can measure the resonance curve of the averaged particle angular response. The location of the resonance peak directly reveals the turbulent vorticity magnitude.

While the present work focuses on understanding particle angular dynamics, it lays essential groundwork for subsequent investigations into how these actively tunable particles can shape turbulent structures, opening possibilities for flexible manipulation of turbulence, where propeller-like particles could be tuned to navigate specific flow structures, inject energy into turbulence, and influence flow properties such as intermittency[55]. More broadly, our study provides fundamental mechanisms that externally tunable rotational dynamics could be leveraged to design active materials[56–58]. If organized coherently, their collective behavior may also exhibit hydrodynamic properties analogous to systems with odd viscosity, which may be promising experimental realizations of chiral fluids[59,60].

This research paves the way for innovative experimental techniques, offering a simple yet underutilized approach to studying and controlling turbulence dynamics.

## Methods
### Experimental setup
The experimental setup comprises three fundamental components: a turbulence generator, a magnetic field generator, and magnetic particles. Detailed specifications of the experimental apparatus, along with a comprehensive description of the particle rotation tracking methodology, are not relevant to the core physics of current work, and hence they are provided in our dedicated instrumentation and methodology paper[61]. Here, we present only the essential information pertinent to the current study.

Turbulence generator: The experiments are conducted in a confined Von Kármán-type turbulent flow, which serves as a model system for studying HIT, particularly in the context of Eulerian and Lagrangian statistics[48,49,62–64].

The working fluid is water, contained within an octagonal-shaped vessel with an internal diameter of $2R = 150$ mm and a height of 220 mm. This specific geometry is chosen to enhance optical accessibility for visualization. The flow is driven by two counter-rotating disks, each with a diameter of 150 mm, separated by a distance of 150 mm, and rotating at the same angular velocity magnitude in opposite directions. Each disk is equipped with six straight blades, each 10 mm in height, designed to enhance inertial stirring. The disks are powered by two calibrated DC motors operating at a constant voltage, ensuring that the angular velocity $\Omega$ remains stable over time with a precision of approximately 1%. The motor rotation frequency can be adjusted within the range of 0.25–8 Hz. Under these conditions, a fully developed turbulent flow with up to $Re_\lambda = 447$ can be generated in the central region of the container with small-scale statistics close to local isotropy[46–49,62–64]. This region, measuring approximately $20 \times 20 \times 20$ mm, serves as the primary domain for measurements. To enable particle rotational motion tracking, a Photron NOVA S6 high-speed camera, operating at 3000 frames per second with a resolution of $1024 \times 1024$ pixels, is employed to record particle motion within the flow field.

Magnetic field generator: A uniform rotating planar magnetic field within a restricted measurement volume, $\mathbf{H} = H/\mathbf{h}$, with an angular frequency $\omega_H$, is generated using a system of two pairs of mutually perpendicular Helmholtz coils. The vertical pair (marked in red in Fig. 1a) has an interior diameter of 300 mm, while the horizontal pair (marked in blue in Fig. 1a) has an interior diameter of 245 mm. For each pair, the separation between the two coils is equal to the corresponding coil diameter.

Each Helmholtz coil pair generates a magnetic field along a single axis. By driving the two perpendicular coil pairs with AC currents that have a phase difference of $\pi/2$, a resultant magnetic field is produced that rotates at a constant angular velocity $\omega_H$. This system is capable of generating rotating magnetic fields with frequencies of up to 50 Hz.

Magnetic particles: The magnetic particles are produced by coating spherical Styrofoam cores with a layer of magnetic paint, which is paramagnetic. The Styrofoam cores are initially arranged in a fixed, evenly spaced configuration on a substrate. The magnetic paint is then applied from the top of the substrate to the Styrofoam spheres through a spraying process, ensuring that all particles are coated in a consistent manner; however, at the single-particle level the coating is not perfectly homogeneous, and small variations in thickness or distribution lead to slight differences in magnetic properties between particles. Finally, the coated particles are detached from the substrate.

The particle volume fraction used in the experiments is 0.17%, ensuring that the system remains in the dilute regime. This low concentration minimizes inter-particle interactions, allowing us to focus on the dynamics of particle response to the complex interplay between turbulent fluctuations and the external magnetic field, without significant effects from collective dynamics.

These particles exhibit anisotropy in three key aspects: (1) Shape: The particles are approximately spherical, with a mean diameter of $0.762 \pm 0.066$ mm. (2) Density: The magnetic particles have a mean

density of $\rho_{particle} = 208 \pm 14$ kg/m³. (3) Magnetic Properties: Due to the non-uniform distribution of the magnetic paint on the particle surface, the magnetic properties vary between particles. Specifically, the magnetic moment magnitude $|\mathbf{m}|$, the orientation of the magnetic moment, and the spatial distribution of magnetization across the surface differ from particle to particle. This variability influences how individual particles respond to an external magnetic field. All these three aspects of particle polydispersity can introduce minor differences between the simulation results and the experimental observations. However, these differences remain small, and overall, the simulations show good agreement with the experiments. More importantly, our theoretical modeling and the simulations successfully capture key physical features observed in the experiments, such as the location of peak responses.

Particle rotation tracking: The particle rotational motion is tracked using an efficient particle rotation tracking algorithm. The algorithm initiates by pre-processing the experimental images through the application of Gaussian filtering to mitigate noise, contrast enhancement to optimize image quality, and central cropping to ensure precise target particle localization. Subsequently, a set of discrete candidate rotation angles is defined, and corresponding theoretical rotated images are generated utilizing 3D rotation matrices. For each temporal frame, the algorithm computes the cross-correlation coefficient between the actual experimental image and each theoretical rotated image to determine the optimal matching rotation angle. By tracking these optimal angles across the time series, the rotational motion trajectory of the particle is reconstructed and visualized. More details of the tracking methodology are reported in our dedicated instrumentation and methodology paper[61].

## Theoretical model

A paramagnetic particle moves in a homogeneous and isotropic turbulent flow while subjected to an external magnetic field $\mathbf{H}$. The magnetic field rotates in the *xoy*-plane (laboratory frame) with an angular velocity $\omega_H$ and intensity $H$, expressed as $\mathbf{H} = H\mathbf{h} = H(\cos(\omega_H t), \sin(\omega_H t), 0)^{\mathsf{T}}$. The particle is modeled as a sphere with intrinsic magnetic anisotropy (same as that of a prolate spheroid of aspect ratio 1.2), characterized by a preferred magnetization direction $\mathbf{n}$, which dynamically evolves over time. This instantaneous unit vector is given by $\mathbf{n} = (\sin\alpha\cos\psi, \sin\alpha\sin\psi, \cos\alpha)^{\mathsf{T}}$, where $\alpha$ and $\psi$ (defined in Fig. 2a) describe the orientation of $\mathbf{n}$.

When the particle is placed in an external magnetic field, $\mathbf{H}$, its induced magnetic moment takes the form[19]

$$\mathbf{m} = \mathcal{V}_p H[\chi_\perp \mathbf{h} + (\chi_\parallel - \chi_\perp)(\mathbf{n} \cdot \mathbf{h})\mathbf{n}], \tag{4}$$

where $\mathcal{V}_p = 4\pi r^3/3$ is the particle volume, with $r$ denoting its radius. The anisotropic magnetic susceptibilities $\chi_{\parallel,\perp}$, arising due to the anisotropy of the particle shape (e.g., spheroidal) and the demagnetizing field factors $N_{\parallel,\perp}$, and $\chi_{\parallel,\perp} = \chi_0/(1 + \chi_0 N_{\parallel,\perp})$[51], causes the particle to respond differently to an external magnetic field depending on the direction. The subscripts parallel ($\parallel$) and perpendicular ($\perp$) refer to directions relative to the preferred magnetization direction $\mathbf{n}$. Due to the tendency of the magnetic moment $\mathbf{m}$ to align with the magnetic field direction $\mathbf{h}$, the rotating magnetic field induces the particle rotational motion. The magnetic torque acting on the particle is given by $\mathbf{T}_M = \mu_w \mathbf{m} \times \mathbf{H}$, where $\mu_w$ is the magnetic permeability. The magnetic torque tries to align the particle's preferred magnetic axis ($\mathbf{n}$) with the magnetic field direction ($\mathbf{h}$) and is balanced by the rotational drag torque (approximated in the Stokes regime) $\mathbf{T}_D = -\xi_r(\boldsymbol{\omega}_p - \boldsymbol{\omega}_f)$[65], where $\xi_r = 8\pi\mu r^3$ is the rotational drag coefficient and $\mu$ is the dynamic viscosity of water.

The particle rotational velocity $\boldsymbol{\omega}_p$ naturally decomposes into two components: a perpendicular component $\boldsymbol{\omega}_{p,\perp}$ (tumbling) and a parallel component $\boldsymbol{\omega}_{p,\parallel}$ (spinning) along $\mathbf{n}$.

The particle rotational motion is governed by the balance of torques, and we consider the overdamped condition[50]

$$\mathbf{T}_M + \mathbf{T}_D = \mathbf{0}, \tag{5}$$

with $\mathbf{T}_M$ the magnetic torque and $\mathbf{T}_D$ the rotational drag torque.

By projecting this balance along and perpendicular to $\mathbf{n}$, we obtain the equations governing tumbling and spinning. The tumbling dynamics, $\dot{\mathbf{n}}$, obey

$$\frac{d\mathbf{n}}{dt} = \omega_a(\mathbf{n} \cdot \mathbf{h})(\mathbf{h} - (\mathbf{n} \cdot \mathbf{h})\mathbf{n}) + \boldsymbol{\omega}_f \times \mathbf{n}, \tag{6}$$

where $\omega_a = \mu \mathcal{V}_p H^2(\chi_\parallel - \chi_\perp)/\xi_r$ is the normalized magnetic field intensity, which is also called anisotropic frequency[19,50], with $\omega_a > 0$ ($\omega_a < 0$) for a prolate (oblate) particle.

The particle spinning rate $\boldsymbol{\omega}_{p,\parallel}$ satisfies

$$\frac{d\left(\xi_r \boldsymbol{\omega}_{p,\parallel}\right)}{dt} = -\xi_r\left(\boldsymbol{\omega}_{p,\parallel} - (\boldsymbol{\omega}_f \cdot \mathbf{n})\mathbf{n}\right), \tag{7}$$

where for the overdamped particle, we have $\frac{d(\xi_r \boldsymbol{\omega}_{p,\parallel})}{dt} = \mathbf{0}$. $\boldsymbol{\omega}_f$ is half of the vorticity of the background flow field, which is obtained through performing Direct Numerical Simulations (DNS) using a pseudo-spectral method. The validity of the overdamped model is further discussed in the Supplementary Information.

The total particle angular velocity is thus $\boldsymbol{\omega}_p = \boldsymbol{\omega}_{p,\parallel} + \boldsymbol{\omega}_{p,\perp} = (\mathbf{n} \cdot \boldsymbol{\omega}_p)\mathbf{n} + \mathbf{n} \times \dot{\mathbf{n}}$, with $\dot{\mathbf{n}} = d\mathbf{n}/dt$ (see Eq. (6)). Equations of motion (6) and (7) are numerically integrated using a third-order Runge–Kutta method.

The governing Eq. (5) can be reformulated in terms of the phase lag angle $\beta$ and the orientation angle $\alpha$, where $\alpha$ denotes the angle between the unit vector $\mathbf{n}$ and the *z*-axis, $\hat{\mathbf{z}}$.

To derive the governing equation for $\beta$, we project Eq. (5) onto $\hat{\mathbf{z}}$-direction, yielding

$$\mathbf{T}_M \cdot \hat{\mathbf{z}} + \mathbf{T}_D \cdot \hat{\mathbf{z}} = 0. \tag{8}$$

Here, the magnetic torque component along $\hat{\mathbf{z}}$ is given by $\mathbf{T}_M \cdot \hat{\mathbf{z}} = \omega_a \xi_r \sin^2\alpha \sin\beta \cos\beta$, while the drag torque contribution is expressed as $\mathbf{T}_D \cdot \hat{\mathbf{z}} = -\xi_r(\omega_{p,z} - \omega_{f,z})$.

Rewriting Eq. (8) explicitly in terms of $\beta$ and $\alpha$, we obtain

$$\dot{\beta} = \omega_H - \frac{\omega_a}{2}\kappa \sin 2\beta - \omega_{f,z}, \tag{9}$$

with the coefficient $\kappa = \sin^2\alpha$.

To establish the evolution equation for $\alpha$, we introduce a unit vector $\mathbf{N} = (-\sin\psi, \cos\psi, 0)^{\mathsf{T}}$, which denotes the direction perpendicular to $\mathbf{n}_{xoy}$, the projection of $\mathbf{n}$ onto the *xoy*-plane (see Fig. 2a for the definition). Projecting Eq. (5) along $\mathbf{N}$ leads to

$$\mathbf{T}_M \cdot \mathbf{N} + \mathbf{T}_D \cdot \mathbf{N} = 0. \tag{10}$$

Here, the magnetic torque component along $\mathbf{N}$ is given by $\mathbf{T}_M \cdot \mathbf{N} = \frac{1}{2}\omega_a \xi_r \cos^2\beta \sin 2\alpha$, while the drag torque contribution is $\mathbf{T}_D \cdot \mathbf{N} = -\xi_r(\boldsymbol{\omega}_p \cdot \mathbf{N} - \boldsymbol{\omega}_f \cdot \mathbf{N})$. Noting that $\boldsymbol{\omega}_p \cdot \mathbf{N} = \dot{\alpha}$, Eq. (10) can be rewritten explicitly as

$$\dot{\alpha} = \boldsymbol{\omega}_f \cdot \mathbf{N} + \frac{\omega_a}{2}\sin 2\alpha \, \cos^2\beta. \tag{11}$$

To capture the essential physics of the system, we assume $\kappa \approx 1$, which is reasonable because $\kappa$ follows a probability density function that is sharply peaked at 1 and decays rapidly to zero for values smaller

than unity within the investigated parameter space (further details are provided in the Supplementary Information). The equation for $\beta$ can be reduced to

$$\dot\beta = \omega_H - \frac{\omega_a}{2}\sin 2\beta - \omega_{f,z}, \tag{12}$$

which is controlled closely by the magnetic field.

Equation (12) is used to predict the phase transition between the phase-locked regime and the turbulent-dynamics and back-and-forth regimes (blue solid line in Fig. 3m), and is ultimately describing the dynamics that gives rise to the mechanism of SR (Fig. 4c–g). Similar dynamical equations as Eq. (12) have been widely applied in multiple physical systems, including condensed matter physics, chemical and biological systems, optics and laser physics, electrical engineering, and financial econometrics[66,67], with a range of related physical problems, such as Josephson junctions[68–72], vortex diffusion and flux-line dynamics in superconductors[68,73,74], reaction-rate characteristics relevant to chemistry, engineering, and biology[75], Brownian motors[76], and decision-making processes under uncertainty which resemble financial market fluctuations[67,72]. The universality of this equation suggests that our findings have broad implications beyond the specific context of magnetic particles in turbulence, offering deeper theoretical insights into nonlinear dynamical systems and facilitating the development of new applications in diverse scientific and technological domains.

Now we show the details of predicting the boundary (Fig. 3m, blue solid line) that separates the phase-locked regime (Fig. 3m, purple region) from the back-and-forth and turbulent-dynamics regimes. The phase-locked regime exists if the equation $\dot\beta = 0$ admits fixed points. Based on the simplified governing equation, Eq. (2) or Eq. (12), this condition translates to ensuring that the following equation has real solutions for any imposed $\omega_H$:

$$\frac{\omega_a}{2}\sin 2\beta = \omega_H - \omega_{f,z}. \tag{13}$$

The left-hand side of Eq. (13) is bounded within $[-\omega_a/2, \omega_a/2]$, while the right-hand side varies typically over $\left[\omega_H - \sqrt{\sigma(\omega_{f,z})}, \omega_H + \sqrt{\sigma(\omega_{f,z})}\right]$, where $\sigma(\omega_{f,z})$ is the variance of $\omega_{f,z}$. For Eq. (13) to have real solutions for any imposed $\omega_H$, the following condition must hold:

$$\omega_H + \sqrt{\sigma(\omega_{f,z})} \le \frac{\omega_a}{2}, \tag{14}$$

which gives

$$\omega_a \ge 2\omega_H + 2\sqrt{\sigma(\omega_{f,z})}. \tag{15}$$

Notice that the standard deviation of $\omega_{f,z}$ is approximated to be of the order of the Kolmogorov frequency scale, i.e., $\sqrt{\sigma(\omega_{f,z})} \sim \omega_\eta$, where $\omega_\eta \sim 1/\tau_\eta$. In our simulations, we observe that $2\sqrt{\sigma(\omega_{f,z})} \sim \omega_\eta$, which justifies this approximation. Based on this, we obtain the final criterion for the phase-locked regime from Eq. (15) as

$$\omega_a \ge 2\omega_H + \omega_\eta. \tag{16}$$

### Direct numerical simulation of the Navier-Stokes equation

To accurately integrate the governing Eqs. (6) and (7), it is essential to correctly incorporate the Lagrangian vorticity signal of the particles $\boldsymbol{\omega}_f$.

We begin by performing DNS of statistically stationary homogeneous isotropic turbulence (HIT) to obtain the Lagrangian vorticity dynamics ($\boldsymbol{\omega}_f$) of small, passively advected particles. The computational domain consists of a cubic box of size $L = 2\pi$ with periodic

boundary conditions. The simulations employ a pseudo-spectral method[52] for spatial discretization and an Adams-Bashforth scheme for temporal integration, with the standard 2/3 dealiasing rule applied to ensure numerical accuracy[53]. The numerical framework has been validated against multiple integration schemes, interpolation methods, and large-scale forcing techniques[77].

The suspended particles are modeled as point-like, dilute tracers that interact with the turbulent flow through hydrodynamic forces alone. Their dynamics are characterized by the Stokes number (St) and the density contrast ($\beta_p$), which describe their response time relative to the fluid and their mass relative to the surrounding medium, respectively. To capture the essential physics while simplifying the problem, we adopt a one-way coupled model, which has been extensively validated in both experimental and numerical studies[9,10,78–82].

The governing equation for particle motion is

$$\ddot{\mathbf{x}}_p = \beta_p \frac{D[\mathbf{u}_f]_O}{Dt} - \frac{1}{\mathrm{St}}\left(\dot{\mathbf{x}}_p - [\mathbf{u}_f]_O\right), \tag{17}$$

where $[\mathbf{u}_f]_O$ denotes the fluid velocity with the subscript O indicating evaluation at the center of the particle, $\mathbf{x}_p$ represents the particle position, and $\mathbf{v}_p = \dot{\mathbf{x}}_p$ is the particle velocity. The Lagrangian vorticity is computed as $\boldsymbol{\omega}_f = \frac{1}{2}[\boldsymbol{\nabla}\times\mathbf{u}_f]_O$. The density contrast is defined as $\beta_p = \frac{3}{1+2\rho_p/\rho_f}$, where $\rho_p/\rho_f$ is the particle-to-fluid density ratio. The Stokes number is given by $\mathrm{St} = d_p^2/(12\beta_p\nu\tau_\eta)$, where $d_p$ is the particle diameter and $\nu$ is the fluid kinematic viscosity.

In this study, we perform three-dimensional DNS at a resolution of $N^3 = 128^3$ with a Taylor-scale Reynolds number $\mathrm{Re}_\lambda \approx 60$[10,52,53,77] and collect 512 independent samples for three particle types: light ($\beta_p \to 3.0$, representing bubbles where $\rho_p \to 0$), neutral ($\beta_p = 1.0$), and heavy ($\beta_p = 0.01$) particles, all with a Stokes number of $\mathrm{St} = 1.0$. This specific Stokes number is chosen because light particles in HIT tend to accumulate preferentially in vortex filaments, leading to strong dynamic localization[7,9,10,83].

A key consideration in extending our results to higher Reynolds number turbulence is the effect of intermittency, particularly the heavy-tailed distribution of vorticity fluctuations in fully developed turbulence. In principle, for very large Reynolds numbers, the intermittency of the small-scale vorticity field should be accounted for, as extreme vorticity events become more pronounced. However, previous studies have shown that while intermittency affects the fine-scale structure of turbulence, its influence on the rotational dynamics of small particles (compared to the Kolmogorov length scale), particularly in the parameter regimes relevant to our study, remains relatively weak, because the intense vorticity events, although frequent, are spatially localized and short-lived; thus, their cumulative effect on particle rotation is averaged out[81,84,85]. Our results and the underlying mechanisms in the main paper provide valuable insights into the fundamental principles governing particle rotation in turbulent flows. Future research could explore the effects of finite-sized particles and beyond the dilute regime.

## Data availability
The data generated in this study have been deposited on Zenodo at https://zenodo.org/records/17076195[86].

## Code availability
The code used for processing the data and generating the figures is publicly available on Zenodo at https://zenodo.org/records/17076195[86] under the CC-BY-4.0 license.

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

## Acknowledgements

We thank F. van Uittert, G. Oerlemans, and J. van der Veen for technical support. This work is supported by the Netherlands Organization for Scientific Research (NWO) through the use of supercomputer facilities (Snellius) under Grant No. 2023.026 received by F.T. This publication is part of the project "Shaping turbulence with smart particles" with Project No. OCENW.GROOT.2019.031 (F.T., H.J.H.C., and R.P.J.K.) of the research program Open Competitie ENW XL, which is (partly) financed by the Dutch Research Council (NWO).

## Author contributions

F.T. designed the research. Z.W. planned and carried out the simulations. F.T., C.W., H.J.H.C., and R.P.J.K. conceived and planned the experiments. C.W. carried out the experiments. Z.W., X.M.d.W., R.B., and F.T. developed the theoretical formalism and contributed to the interpretation of the results. Z.W. took the lead in writing the manuscript. All authors contributed to and approved the final version of the manuscript.

## Competing interests

The authors declare no competing interests.
