## [Transparent Peer Review file · Nature Communications]

Stochastic resonance of rotating particles in turbulence

Corresponding Author: Professor Federico Toschi

Version 0:

Reviewer comments:

Reviewer #1

(Remarks to the Author)

The article presents a compelling exploration of the rotational dynamics of light magnetic particles in turbulent flows under the influence of a rotating magnetic field. By combining experimental studies in a von Kármán turbulent flow, theoretical modeling, and direct numerical simulations (DNS) of homogeneous isotropic turbulence using a point-particle approximation, the authors deliver novel insights into the interplay between turbulence and magnetic forcing.

Three main contributions, which I find particularly novel and impactful, stand out:

- The first experimental observation of stochastic resonance (SR) in turbulent vorticity, offering a new mechanism for enhancing signal detection in noisy environments.
- The identification of a symmetry-breaking phenomenon in the particles' rotational response under zero-mean magnetic forcing, revealing a rich nonlinear dynamical behavior.
- The proposal of a metrological technique for probing turbulent vorticity, using magnetic resonance of embedded particles, potentially valuable in microfluidic and rheological applications.

These findings are, in my opinion, not only important for the fundamental understanding of turbulence but also open new avenues in active microrheology, chiral fluid design, and metrological techniques. The demonstrated "surfing effect" highlights the nonlinear coupling between deterministic and stochastic forces, and the introduced phase-transition framework could inspire broader applications beyond turbulence research.

That said, several technical clarifications and improvements in presentation would strengthen the manuscript and ensure reproducibility. I believe the article has strong potential for publication in Nature Communications, and I will be happy to give a final recommendation after the authors address the points detailed below.

1. Turbulence Generator Details

The description of the experimental facility is lacking. The authors should provide key flow characteristics, including the Reynolds number, energy injection/dissipation scales, and relevant integral/Kolmogorov timescales.

2. Particle Characterization

Additional detail on the particles is necessary. Specifically:

- What is their size relative to the turbulence length scales?
- Can they be reasonably treated as point particles, or must finite-size effects be considered?
- Quantify diverse sources of polydispersity and provide an estimate of the magnetic susceptibility.
- The roughness appears significant from images, has this been measured or characterized?

3. Magnetic Field Details

While magnetic field amplitudes are mentioned in figure captions, they should also be clearly specified in the Methods section, along with the typical frequency range used.

4. Rotational Tracking Methodology

Please elaborate on the method used for tracking rotational motion:

- What pattern or texture is exploited for angular tracking?
- Is any post-processing or filtering applied?
- What is the temporal resolution and signal-to-noise ratio on angular velocity. Is it sufficient to eventually hope extracting

rotational accelerations for future studies?

5. Rotational Dynamics Modeling

The authors rely on Stokes rotational drag. Given the aspect ratio ~ 1.2 , have they estimated the role of inertial torques? These can be non-negligible even for weakly anisotropic, sub-Kolmogorov particles.

6. Validity of the Point Particle Approximation

How appropriate is the point-particle model used in simulations, considering the actual particle size in experiments?

7. Neglecting Rotation-Translation Coupling

The assumption of decoupled translational and rotational dynamics may not hold for finite-size or anisotropic particles. Could the authors assess the possible impact of these couplings?

8. Overdamped Approximation

The overdamped limit is central to the analysis. The authors should discuss its validity more thoroughly:

- Under what conditions (e.g. magnetic field strength, frequency, particle size/moment inertia/magnetization) is the approximation justified?
- Have they evaluated rotational acceleration statistics to confirm the absence of inertial effects beyond first moments?

9. Effect of von Kármán Flow Properties

While the von Kármán setup is a standard turbulence generator, it is not isotropic or homogeneous, particularly due to its central core with non-zero mean helicity. Could this large-scale anisotropy influence the observed symmetry breaking or other phenomena? A discussion on this point would improve the interpretation of results and their generalizability.

Reviewer #2

(Remarks to the Author)

The authors report on the rotational dynamics of a magnetic particle (subjected to a rotating magnetic field) in a turbulent von Kármán flow (generated by two counter-rotating impellers). They observe three particle-rotation regimes experimentally and numerically: A phase-locked regime synchronized with the field rotation, a back-and-forth particle motion, and a turbulence-dominated one. The main result of the manuscript concerns the numerical evidence of stochastic resonance of the particle angle occurring for intermediate stochastic noise of turbulence.

To do so, the authors built an elegant experimental system and developed a numerical method to probe the stochastic resonance. Although redundancies and back-and-forth are too frequent, the manuscript is rather well written, and the figures are presented very cleanly. However, I find that the experimental part is too reduced and concerns only the phase-locked regime and the back-and-forth regimes, which are not new in the literature, the latter being not sufficiently quoted. Moreover, I have several serious concerns about the claims made by the authors.

Major comments:

1) The synchronized regime and the back-and-forth regime are known phenomena occurring without noise, as already observed experimentally for a compass (V. Croquette et al., J. Phys. Lett. 1981, A. Poyé et al., Phys. Scr. 2019) or a rolling particle (A. Kaiser et al., Sci. Adv. 2017) in an alternating magnetic field. On the other hand, the turbulent regime occurs when the effect of the magnetic field is negligible, which, in practice, corresponds to a nonmagnetic particle in a turbulent flow. Thus, only the results related to the intermediate noise level are new, including the stochastic resonance. It should be clearly stated in the manuscript, and these references should at least be discussed. The bibliography also fails to include key articles relevant to the topic of this manuscript (see below).

2) The authors want to address an important question: Understanding the coupling between a particle's rotation and the underlying vorticity field in a turbulent flow. The authors state that magnetic particles can control the vorticity of the flow. In turbulent flows, energy is generally injected at large scales (e.g., by an impeller) and cascades towards small scales. In this system studied by the authors, the energy contained at the particle size is therefore much smaller than the energy-containing range, and the particle rotation will surely influence only slightly the dynamics of the flow and its vorticity. The authors could comment on that point.

Such claims could have been supported by experimental measurements in the flow velocity field, but the manuscript appears to make limited use of the experimental facility. Most of the results are numerical simulations, in particular, the main result of stochastic resonance in turbulence, which I believe could have also been evidenced experimentally. Consequently, the significance of this work to the field is questionable, mainly due to the lack of experimental data to compare with the numerical resolution of a single particle equation. Reading the abstract, I would have expected more experimental results (only Figs. 1 and 2 are experimental), and the main result of the manuscript is numerical. Moreover, the authors use optical methods to track the particle and measure the flow vorticity. I believe measurements with PTV or PIV would also give very high-quality vorticity data and probably be less intrusive to the flow than using particles with a size above the Kolmogorov length scale.

3) The main message of the manuscript is that vorticity fluctuation effects on a magnetic particle can be seen as "noise" without the need to solve complex hydrodynamics equations. However, I do not see any results showing the impact of the

particle's rotation on the flow via hydrodynamic interactions, which is stated in the introduction as a mechanism to control turbulence.

4) There is a claim in the abstract, introduction, and conclusion that using magnetic particles to probe turbulent fluctuations can circumvent optical techniques. However, the experimental results in the manuscript solely rely on optical measurements. Can the author clarify what they have in mind?

5) How do the authors measure the Kolmogorov timescale in their experimental setup, presented in Fig. 2? What is the experimental Stokes' number?

6) The title is misleading and should be explicit, something like "Stochastic resonance dynamics of rotational magnetic particles in turbulence".

7) What is the Reynolds number? It should be clearly stated, as is standard in any turbulence study. Depending on its value, the rotational drag coefficient could be laminar (as used in the manuscript) or turbulent. Thus, it could change the conclusion of the manuscript. It should be relatively easy numerically to take this point into account.

8) Why experimental PDFs are skewed in Figs. 2b-d, whereas the signals in Fig. 1g and Fig.1i look symmetric?

9) It would have been interesting to include more experimental points in Fig. 3 (beyond the three already provided) to perform a reasonable comparison with simulations.

10) The typical system size and magnetic field strength should appear in the main text (instead of the Methods Section). Generally speaking, the frequent back-and-forth between the main text, Methods section, and Supplemental Information disrupts the reading flow. Some redundancies also appear in these different parts.

11) I believe there is a mistake in the magnetic particle density line 489. 0.2 kg/m^3 is a density 6 times lighter than air, please correct that. Why not choose neutrally buoyant particles instead of very light ones compared to water?

12) The bibliography is extensive (73 references). However, several experimental studies, conducted in different research groups, have investigated 3D turbulence generated by magnetic particles (e.g., E. Falcon et al. PRL 2017, A. Cazaubiel et al. Phys. Rev. Fluids 2021, J.B. Gorce et al. Phys. Rev. Lett. 2024), or regimes reminiscent of 2D turbulence (e.g., N. Francois et al. Phys. Rev. X 2014, G. Kokot et al. PNAS 2017, M. Bourgoin et al. Phys. Rev. X 2020). Including some of these works in the bibliography would strengthen the contextual background.

13) Is the magnetic coating uniform? There is an inconsistency on this point in the Methods section (lines 476 and lines 490-491)? What determines the direction of the magnetic moment of each particle?

14) Is it in Nature Communications' policies to publish the dedicated instrumentation and methodology in another journal, as indicated in lines 426-427?

Minor comments

In line 489 and the Supplemental Information, the accuracy of the magnetic particle diameter is in micrometers. Is it significant?

Movies 6, 7, and 8 are long (5 min each) and could be sped up.

Reviewer #3

(Remarks to the Author)

Version 1:

Reviewer comments:

Reviewer #1

(Remarks to the Author)

I have now carefully read the responses of the authors to the points I raised in my previous review, as well as the new version of their manuscript. They both clearly address all the remarks I pointed and substantially improve in my opinion the clarity and robustness of the study.

My recommendation at this point is to accept the article for publication in Nature Communications.

Reviewer #2

(Remarks to the Author)

The author considered most of my suggestions, including adding experimental data, changing the title, and focusing the scope of the study on stochastic resonance, which is the main result, rather than the phase-locked or back-and-forth regimes already explored. Moreover, the authors have added more experimental details, but some comments are still lacking.

The revised manuscript has been improved and is now ready for publication in Nature Communications, provided the authors address the points outlined below.

i) In contrast to the response given to my Comment 2, the new experimental points performed do not appear in Fig. 3(m) as in Fig. R1 for Reviewer 2. I think it is just an omission.

ii) The authors still claim that this concept has strong analogies with magnetometry techniques used in biomedical applications, but do not give any references or explain how they can scale up this technique.

iii) The Stokes number asked in my Comment 5 is still not given.

iv) The sentence, answering my Comment 11, “to this end, magnetic particles are designed to be light and of a size comparable to the Kolmogorov scale, allowing them to preferentially enter and interact with vortex filaments” is an important statement that should be clearly emphasized in the manuscript.

v) The Comment 14 is still not answered: Is it in Nature Communications’ policies to publish the dedicated instrumentation and methodology of this manuscript in another journal – see new Ref. [60] (experimental) and Ref. [54] (numerics) of the same authors’ group? Furthermore, the reliance on arXiv publications posted after the first round of review raises concerns about the transparency of the revision process.

Minor:

In the Theoretical model Section, the sentence “and details can be found later” should be removed.

Reviewer #3

(Remarks to the Author)

Version 2:

Reviewer comments:

Reviewer #2

(Remarks to the Author)

The author considered all my suggestions.

The revised manuscript is now ready for publication in Nature Communications.

Reviewer #3

(Remarks to the Author)

Manuscript: Stochastic resonance of rotating particles in turbulence

Tracking: NCOMMS-25-34227-T

Authors: Ziqi Wang, Xander M. de Wit, Roberto Benzi,
Chunlai Wu, Rudie P. J. Kunnen, Herman J. H. Clercx,
and Federico Toschi

September 5, 2025

Response to Reviewer 1

We would like to thank the reviewer for the thorough reading of our manuscript, for his/her many insightful comments, and suggestions that allowed us to considerably improve our manuscript. We really appreciate these efforts by the reviewer.

Synopsis: *The article presents a compelling exploration of the rotational dynamics of light magnetic particles in turbulent flows under the influence of a rotating magnetic field. By combining experimental studies in a Von Kármán turbulent flow, theoretical modeling, and direct numerical simulations (DNS) of homogeneous isotropic turbulence using a point-particle approximation, the authors deliver novel insights into the interplay between turbulence and magnetic forcing.*

Three main contributions, which I find particularly novel and impactful, stand out:

- *The first experimental observation of stochastic resonance (SR) in turbulent vorticity, offering a new mechanism for enhancing signal detection in noisy environments.*
- *The identification of a symmetry-breaking phenomenon in the particles' rotational response under zero-mean magnetic forcing, revealing a rich nonlinear dynamical behavior.*
- *The proposal of a metrological technique for probing turbulent vorticity, using magnetic resonance of embedded particles, potentially valuable in microfluidic and rheological applications.*

These findings are, in my opinion, not only important for the fundamental understanding of turbulence but also open new avenues in active microrheology, chiral fluid design, and metrological techniques. The demonstrated “surfing effect” highlights the nonlinear coupling between deterministic and stochastic forces, and the introduced phase-transition framework could inspire broader applications beyond turbulence research.

That said, several technical clarifications and improvements in presentation would strengthen the manuscript

and ensure reproducibility. I believe the article has strong potential for publication in Nature Communications, and I will be happy to give a final recommendation after the authors address the points detailed below.

Response: We are very grateful to the reviewer for all the suggestions on how to improve our manuscript. In line with the comments of the reviewer, we have made a sincere and dedicated effort for the revision of the manuscript. In addition, a point-by-point response to all comments/suggestions is provided below. Please note that all modifications made in the revised manuscript and Supplementary Material based on the comments of the reviewer are highlighted in blue. For convenience, here we have also quoted the modified content in the revised main paper and the revised Supplementary Material and highlighted them in blue. In the following, we answer/clarify all points raised by the reviewer.

Comment 1: *Turbulence Generator Details*

The description of the experimental facility is lacking. The authors should provide key flow characteristics, including the Reynolds number, energy injection/dissipation scales, and relevant integral/Kolmogorov timescales.

Response: In the following, we provide a description of the key flow characteristics relevant to our experiments.

First, we note that the detailed description of the experimental setup, –including the turbulence generator (a modified French washing machine), the magnetic field generator (Helmholtz coils), the magnetic particles, and the integrated design of the facility– has been thoroughly reported and discussed in a companion paper (Wu et al., 2025), focusing on the experimental design.

For clarity and completeness in the present work, we now include the essential flow characteristics, which are directly relevant to the phenomena investigated here.

The measured dissipation rate ε is determined from the driving torque on the two impellers as $\varepsilon = T\Omega/(\rho V_w)$ (Mordant, 1997; Labbé et al., 1996), where T is the sum of the driving torques from the impellers at the top and bottom of the water tank and $V_w = 2.8 \times 10^{-3} \text{ m}^3$ is the volume of the water tank. The driving torques are measured using two strain gauge torque meters mounted on the motors.

In our experiments, we investigated two turbulence intensities: (1) the weak-turbulence condition, for which the results are shown in Fig. 2(b–d) of the main text; and (2) the strong-turbulence condition, for which the results are presented in Supplementary Fig. 1(b).

For the weak-turbulence case, the impellers rotate at 0.83 Hz. The corresponding mean energy dissipation rate is $\varepsilon = 0.037 \pm 0.002 \text{ m}^2/\text{s}^3$, which yields a Taylor-scale Reynolds number of $Re_\lambda = 398$, a Kolmogorov length scale of $\eta = 0.072 \text{ mm}$, and a Kolmogorov time scale of $\tau_\eta = 5.20 \text{ ms}$.

For the strong-turbulence case, the impellers rotate at 1.21 Hz. The corresponding mean energy dissipation rate is $\varepsilon = 0.075 \pm 0.001 \text{ m}^2/\text{s}^3$, which yields a Taylor-scale Reynolds number of $Re_\lambda = 447$, a Kolmogorov length scale of $\eta = 0.061 \text{ mm}$, and a Kolmogorov time scale of $\tau_\eta = 3.66 \text{ ms}$.

We have clarified these aspects in the revised Supplementary Material.

The modifications in the Supplementary Material are listed below:

The measured dissipation rate ε is determined from the driving torque on the two impellers as $\varepsilon = T\Omega/(\rho V_w)$ (Mordant, 1997; Labbé et al., 1996), where T is the sum of the driving torques from the impellers at the top and bottom of the water tank and $V_w = 2.8 \times 10^{-3} \text{ m}^3$ is the volume of the water tank. The driving torques are measured using two strain gauge torque meters mounted on the motors. In our experiments, we investigated two turbulence intensities: (1) the weak-turbulence condition, for which the results are shown in Fig. 2(b–d) of the main text; and (2) the strong-turbulence condition, for which the results are presented in Supplementary Fig. 1(b). For the weak-turbulence case, the impellers rotate at 0.83 Hz. The corresponding mean energy dissipation rate is $\varepsilon = 0.037 \pm 0.002 \text{ m}^2/\text{s}^3$, which yields a Taylor-scale Reynolds number of

$Re_\lambda = 398$, a Kolmogorov length scale of $\eta = 0.072$ mm, and a Kolmogorov time scale of $\tau_\eta = 5.20$ ms. For the strong-turbulence case, the impellers rotate at 1.21 Hz. The corresponding mean energy dissipation rate is $\varepsilon = 0.075 \pm 0.001$ m²/s³, which yields a Taylor-scale Reynolds number of $Re_\lambda = 447$, a Kolmogorov length scale of $\eta = 0.061$ mm, and a Kolmogorov time scale of $\tau_\eta = 3.66$ ms.

Comment 2: *Particle Characterization: Additional detail on the particles is necessary. Specifically: What is their size relative to the turbulence length scales? Can they be reasonably treated as point particles, or must finite-size effects be considered? Quantify diverse sources of polydispersity and provide an estimate of the magnetic susceptibility. The roughness appears significant from images, has this been measured or characterized?*

Response: Below, we provide the point-by-point explanation.

- The particles used in our experiments possess three key characteristics that enable the study of their rotational motion under the combined influence of turbulence and magnetic interactions with the rotating magnetic field. The particle diameter is in the range between the Kolmogorov scale and the Taylor microscale, with the radius $r \approx 5\eta$, which ensures interaction with the smallest turbulent flow structures.

- Regarding the point-particle assumption, the particles cannot be strictly treated as point-like because their size is larger than the Kolmogorov length scale, η . However, they are still much smaller than the Taylor microscale. As such, they reside in an intermediate regime where the dominant interaction is with dissipative-scale turbulence, and finite-size effects are limited. Our analysis primarily focuses on the rotational dynamics of the particle, which are more sensitive to the externally applied magnetic torque and local velocity gradients than to large-scale finite-size wake effects. Thus, while the point-particle approximation is not strictly valid, the qualitative conclusions of our study, and most crucially the conception of the stochastic resonance mechanism, remain robust.

- The magnetic susceptibility and anisotropy of our particles are statistically characterized using a Superconducting Quantum Interference Device (SQUID) magnetometer manufactured by Quantum Design Inc. The mean volume magnetic susceptibility $\bar{\chi}_V$ is determined via batch measurements of 20 particles randomly oriented in the sample holder, yielding a mean value of $|\bar{\chi}_V| = 0.136 \pm 0.046$. Magnetic anisotropy was evaluated through single-particle measurements, yielding an average value of $\Delta\chi = 0.011 \pm 0.002$. A magnetic field of 1 T was first applied to allow the particle to freely rotate and align its magnetic moment \mathbf{m}_p with the field \mathbf{B} . Once aligned, the hysteresis loop was recorded along this direction to determine the parallel susceptibility χ_{\parallel} . The perpendicular susceptibility χ_{\perp} was then measured by applying the magnetic field orthogonal to the aligned magnetic moment. A more detailed description of the particles' magnetic properties can be found in our experimental paper, referenced as (Wu et al., 2025). Sources of polydispersity include variation in particle diameter, density, and magnetic susceptibility. Based on imaging of a batch of 50 randomly selected particles, the diameter distribution has a mean of $d = 0.762$ mm with a standard deviation of ± 0.066 mm, corresponding to approximately 8.7% relative polydispersity. The density is $\rho_p = 0.208$ kg/m³ with a variation below 0.014% across the sampled batch. The spread in magnetic susceptibility values has already been quantified above. Taken together, these sources of polydispersity are moderate and do not qualitatively affect the rotational dynamics and the stochastic resonance behavior we report.

- The particles appear “rough” in optical images, which originates from their manufacturing process. Each particle consists of a lightweight polystyrene core coated with a thin layer of magnetic paint containing iron powder. During manual spraying, variations in paint distribution, the side of the particle in contact with the adhesive tape, and minor inconsistencies in the spraying angle and amount result in a non-uniform coating. This process imparts the magnetic properties and anisotropy necessary for particle rotation under the applied magnetic field, and also generates surface patterns that visually resemble “roughness”. However, this apparent roughness is primarily due to the contrast between the bright polystyrene core and the darker magnetic coating, rather than actual geometric protrusions. On the macroscopic scale relevant to particle rotational dynamics, the particle surfaces remain closely spherical. Therefore, we expect that this small roughness does not significantly increase translational drag or rotational resistance beyond the effects already captured by particle size and magnetic torque.

We have clarified these aspects in the revised Supplementary Material.

The modifications in the main paper are listed below:

The particles are designed to probe rotational dynamics under the combined influence of turbulence and a rotating magnetic field. Their size lies between the Kolmogorov and Taylor microscales, with $r \approx 5\eta$, ensuring interactions with the smallest turbulent structures while remaining much smaller than the integral scale and Taylor microscale. Consequently, they cannot be regarded as point-like; however, finite-size effects on the rotation are limited, and the dominant particle–flow interactions occur at dissipative scales. The particles’ magnetic properties were characterized using a Superconducting Quantum Interference Device (SQUID) magnetometer (Quantum Design Inc.). Batch measurements yielded a mean volume susceptibility of $|\bar{\chi}_V| = 0.136 \pm 0.046$, while single-particle measurements revealed an average anisotropy of $\Delta\chi = 0.011 \pm 0.002$.

In optical images as shown in Fig. 1, the particle surfaces may appear “rough.” This appearance originates from the contrast between the bright Styrofoam core and the darker magnetic coating, rather than actual geometric protrusions. Variations in paint deposition, the side of the particle in contact with the substrate during spraying, and small inconsistencies in the magnetic paint spraying angle result in non-uniform coating patterns. These patterns enhance optical contrast, creating the visual impression of surface roughness. However, the actual geometric irregularities are negligible compared to the particle diameter, and the surfaces remain effectively smooth on the hydrodynamic scale. Therefore, while such patterns may introduce minor perturbations to local flow, they do not significantly alter translational drag or rotational resistance beyond the effects already captured by particle size and magnetic torque.

Comment 3: *Magnetic Field Details*

While magnetic field amplitudes are mentioned in figure captions, they should also be clearly specified in the Methods section, along with the typical frequency range used.

Response: The rotating planar magnetic field is calibrated using a SENIS F3A magnetic-field-to-voltage transducer equipped with a fully integrated 3D Hall probe. Within the frequency range of 1 to 50 Hz, the maximum magnetic flux density remains stable at 1.60 ± 0.02 mT. The frequency is set to be 20 Hz in Fig.2 of the main text, and is set to be 10 Hz in Fig. 4 of the main text, which shows the stochastic resonance. Detailed characterization of the magnetic field can also be found in our experimental paper (Wu et al., 2025).

The modifications in the Method section of the main paper are listed below:

The rotating planar magnetic field was calibrated using a SENIS F3A magnetic-field-to-voltage transducer with an integrated 3D Hall probe. Across the frequency range of 1–50 Hz, the maximum magnetic flux density remained stable at 1.60 ± 0.02 mT. The driving frequency is set to be 20 Hz for Fig. 2 and 10 Hz for Fig. 4, which illustrates the stochastic resonance. Further details of the field characterization are reported in our experimental study.

Comment 4: *Rotational Tracking Methodology*

Please elaborate on the method used for tracking rotational motion:

What pattern or texture is exploited for angular tracking? Is any post-processing or filtering applied? What is the temporal resolution and signal-to-noise ratio on angular velocity. Is it sufficient to eventually hope extracting rotational accelerations for future studies?

Response: The magnetic particles are made of Styrofoam cores coated with purple magnetic paint. The original white color of the Styrofoam combined with the purple magnetic paint forms the non-uniform surface pattern. This random pattern serves as the feature for angular tracking, as the algorithm computes rotational motion by analyzing the changes in the texture across frames. The distribution of magnetic paint is not easily controllable due to the manual fabrication process: we characterize the particle magnetism statistically. To reduce image noise, a Gaussian filter is applied to each frame before tracking. The kernel size is set to 3 pixels in our experiments, which was found sufficient to smooth minor fluctuations without affecting the rotational features. A more detailed description and validation of the rotational tracking methodology are reported in our experiment focused work under the section of “IV. IMAGE PROCESSING AND PARTICLE ROTATION TRACKING” (Wu et al., 2025). The frame rate of our recordings is 3000 fps, corresponding to a temporal resolution of 0.33 ms. In validation experiments conducted to implement this rotation tracking algorithm on magnetic particles (Wu et al., 2025), the signal-to-noise ratio (SNR) of the measured angular velocity (normalized by the reference angular velocity of the magnetic field) was evaluated using the definition $SNR = 20 \log_{10}(\mu/\sigma)$, where μ and σ denote the mean and standard deviation of the normalized angular velocity, respectively. The SNR was found to range from 16.0 to 37.0 as the reference angular velocity increased. For rotation frequencies above 20 Hz, the SNR exceeded 30, indicating that rotational accelerations can be reliably extracted when particles rotate at frequencies higher than 20 Hz. In turbulent flows, particle rotation can be more irregular than in controlled conditions; while our current results demonstrate that rotational velocities and accelerations can be reliably measured under well-characterized rotations, further investigation is needed to fully assess the method performance especially in highly turbulent environments.

Comment 5: *Rotational Dynamics Modeling*

The authors rely on Stokes rotational drag. Given the aspect ratio 1.2, have they estimated the role of inertial torques? These can be non-negligible even for weakly anisotropic, sub-Kolmogorov particles.

Response: First, we clarify that the particles in the simulation is modeled as a sphere. To enable these spherical particles experience a magnetic torque in a rotating magnetic field, we assume that the particles possesses magnetic properties, specifically anisotropic susceptibilities χ_{\parallel} and χ_{\perp} . These susceptibilities are set to be identical to those of a prolate spheroid with an aspect ratio 1.2, since χ_{\parallel} and χ_{\perp} are uniquely determined by the aspect ratio of the particle (Osborn, 1945). This specific aspect ratio is chosen based on experimental estimations of the magnetic particle properties, aiming to closely mimic the behavior of particles observed in experiments for a direct one-to-one comparison between simulation and experimental results.

Secondly, we agree that a careful assessment of inertial effects is crucial for the model validity. Our rotational dynamics model relies on the overdamped condition, which assumes that inertial torques are negligible compared to viscous and magnetic torques. To justify this assumption, we now estimate the particle rotational response time (τ_{rot}) and compared it to the characteristic timescale of the fluid vorticity (τ_{η}).

Based on our experimental parameters:

- Particle diameter (mean): $d = 0.762$ mm (radius $r = 0.381$ mm)
- Particle density (mean): $\rho_p = 208$ kg/m³
- Fluid viscosity (water at room temp): $\mu \approx 1.0 \times 10^{-3}$ Pa · s
- Characteristic timescale of fluid vorticity: $\tau_{\eta} = 7.56$ ms

For a spherical particle, the rotational response time can be estimated as $\tau_{\text{rot}} = \frac{I}{\xi_r}$, where $I = \frac{2}{5}\mathcal{V}_p\rho_p r^2$ is the particle moment of inertia, and $\xi_r = 8\pi\mu r^3$ is the rotational drag coefficient. So $\tau_{\text{rot}} = \frac{\rho_p r^2}{15\mu}$.

Substituting the values:

$$\begin{aligned}\tau_{\text{rot}} &= \frac{208 \text{ kg/m}^3 \times (0.381 \times 10^{-3} \text{ m})^2}{15 \times 1.0 \times 10^{-3} \text{ Pa} \cdot \text{s}} \\ &= 2.01 \times 10^{-3} \text{ s} = 2.01 \text{ ms}.\end{aligned}$$

Now, comparing the two timescales:

- Particle rotational response time: $\tau_{\text{rot}} = 2.01$ ms.
- Fluid vorticity timescale: $\tau_{\eta} = 7.56$ ms.

We observe that τ_{rot} is approximately $2.01/7.56 = 0.266$ or about one-quarter of τ_{η} , indicating that the particle rotational dynamics are significantly faster than the characteristic changes in the fluid vorticity.

This suggests that the particle can respond quickly to the fluid rotational fields, making the assumption of an overdamped regime a reasonable first approximation. Furthermore, many novel phenomena of magnetic particles have been found in an overdamped system, e.g., colloidal chains, microscopic rods, flexible filaments, or dynamical re-orientations (Melle et al., 2002; Qian et al., 2008; Belovs and Cēbers, 2006; Dhar et al., 2007; Tierno et al., 2009).

Of course the magnetic particles used in our experiments may display a slight anisotropy, the derivation of τ_{rot} for an equivalent sphere provides a strong first-order estimate. For weakly anisotropic particles and in situations where the particle response time is considerably shorter than the characteristic fluid scales, inertial torques are often considered negligible. In such cases, the particle angular response is largely dominated by drag torque rather than inertial ones. Moreover, the good agreement observed between our experimental results and numerical simulations provides strong direct evidence that inertial effects are negligible in the regimes studied.

Therefore, based on this quantitative comparison of timescales, we maintain that the overdamped model provides a robust framework for analyzing the rotational dynamics in this study. We acknowledge that for cases with significantly larger τ_{rot} or specific highly dynamic flow conditions, the inclusion of inertial torques would indeed be necessary, but this is beyond the scope of the present research.

We have clarified this careful assessment of inertial effect to justify the overdamped model validity in the revised manuscript.

The modifications in the SI of the paper are listed below:

To justify the validity of the overdamped model, we carefully assessed inertial effects. For a spherical particle, the rotational response time can be estimated as $\tau_{\text{rot}} = \frac{I}{\xi_r}$, where $I = \frac{2}{5}\mathcal{V}_p\rho_p r^2$ is the particle moment of inertia, and $\xi_r = 8\pi\mu r^3$ is the rotational drag coefficient. Substituting these into the equation for τ_{rot} simplifies to $\tau_{\text{rot}} = \frac{\rho_p r^2}{15\mu}$. Using our experimental parameters of $r_p = 0.381$ mm, $\rho_p = 208$ kg/m³, $\mu \approx 1.0 \times 10^{-3}$ Pa · s, and $\tau_\eta = 7.56$ ms. The rotational response time is $\tau_{\text{rot}} = 2.01$ ms. We observe that τ_{rot} is approximately $2.01/7.56 = 0.266$ or about one-quarter of τ_η , indicating that the particle rotational dynamics are significantly faster than the characteristic changes in the flow vorticity. Therefore, the particle can respond quickly to the flow vorticity, making the assumption of an overdamped regime a reasonable first approximation. Note that our experimental frequencies are chosen to ensure the particle can follow the driving field within the overdamped regime, i.e., the frequency of the magnetic field ω_H is slower than $1/\tau_{\text{rot}} \approx 2000$ Hz, so that the particle rotational motion will remain in the overdamped regime.

Comment 6: *Validity of the Point Particle Approximation*

How appropriate is the point-particle model used in simulations, considering the actual particle size in experiments?

Response: The point-particle approximation offers a leading-order description of particle–fluid interaction by neglecting finite-size hydrodynamic effects, and it has been widely adopted in studies of particle-laden flows and particle–field interactions (Maxey and Riley, 1983; Toschi and Bodenschatz, 2009).

In our system, the particle radius is comparable to the Kolmogorov length scale of the turbulence ($r \sim \eta$), placing the particle near the lower end of the inertial range. Under such conditions, the velocity gradients across the particle remain small, allowing the flow to be treated as locally linear. As a result, the rotational dynamics of the particle are primarily governed by the local vorticity rather than by finite-size-induced flow disturbances. This provides a physically sound justification for employing the point-particle model in our numerical simulations.

Furthermore, the dominant driving mechanism for particle rotation in our system includes a well-characterized external magnetic torque, which acts directly on the particle magnetic moment rather than through hydrodynamic interactions. This reduces the sensitivity of the results to finite-size corrections.

Importantly, the suspension used in the experiments is in the dilute regime, with a particle volume fraction of $\Phi_V = 0.17\%$. Such a low concentration results in the fact that the average inter-particle distance is much greater than the particle diameter and thus minimizes particle-particle interactions, allowing us to neglect hydrodynamic interactions between particles. This also justifies the one-way coupling situation in our system (Balachandar and Eaton, 2010), where the flow affects particle motion, but the back-reaction of particles on the flow field is neglected, which further simplifies the problem to a point-particle approach.

Finally, we have validated the simulation results against experimental measurements of particle angular velocity, and observed good agreement within the bounds of experimental uncertainty. This empirical consistency supports the qualitative and semi-quantitative validity of the point-particle approximation in our system.

We have comment on this in the revised manuscript.

The modifications in the Supplementary Information are listed below:

Despite neglecting finite-size hydrodynamic effects, the point-particle approximation offers a leading-order description of particle–fluid interaction and is widely adopted in studies of particle-laden flows (Maxey and Riley, 1983; Toschi and Bodenschatz, 2009). In our system, the particle radius is comparable to the Kolmogorov length scale ($r \sim \eta$), ensuring locally linear flow gradients that primarily govern rotational dynamics via local vorticity. This, coupled with a dominant external magnetic torque and a dilute suspension (volume fraction of $\Phi_V = 0.17\%$ that minimizes particle-particle interactions (Balachandar and Eaton, 2010)), provides a robust physical justification for our model. Crucially, the good agreement between our simulations and experimental measurements of particle angular velocity further validates the point-particle

approximation for this system.

Comment 7: *Neglecting Rotation-Translation Coupling*

The assumption of decoupled translational and rotational dynamics may not hold for finite-size or anisotropic particles. Could the authors assess the possible impact of these couplings?

Response: The particles used in our experiments/simulations are sufficiently small and nearly spherical. Moreover, their mass distribution is approximately uniform. These factors minimize the offset between the center of mass and the hydrodynamic center (center of resistance), reducing the torque generated by translational motion.

This is consistent with previous research efforts. For example, it has been found experimentally (Zimmermann et al., 2011) that the linear and angular velocities and accelerations reveal an intermittency phenomena for the particles whose size lies in the inertial range, close to the integral scale of the underlying turbulent flow. This is inherently a finite size effect; But for particles with very small diameters (compared to the Kolmogorov length scale) the translational and rotational dynamics are not coupled.

Even if weak coupling exists, we note that the main results of our study – such as the emergence of stochastic resonance and the phase lag behavior – are primarily determined by the balance between external torque and rotational viscous drag. Therefore, any minor translation-rotation coupling would not qualitatively alter our conclusions and the revealed mechanisms.

While we assume decoupled translational and rotational dynamics based on particle properties and statistical verification, we agree that it is worthwhile to consider how such coupling, if present, might influence rotational dynamics of the magnetic particles. For particles with non-uniform mass distribution or shape asymmetry, translational motion through a shear or turbulent flow can generate additional hydrodynamic torques due to a mismatch between the center of mass (CM) and the center of resistance (CR). This torque effectively acts as an additional noise or deterministic bias on the rotational motion.

Below we estimate the magnitude of this effect in our system.

We consider a particle with radius $r = 0.381$ mm and a possible CM–CR offset $\delta \sim 0.1r = 0.0381$ mm. The translational velocity at the Kolmogorov scale is approximately

$$v \approx v_\eta \approx \frac{\eta}{\tau_\eta} = \frac{8.7 \times 10^{-5} \text{ m}}{7.56 \times 10^{-3} \text{ s}} \approx 0.0115 \text{ m/s.} \quad (1)$$

The torque induced by translational drag is then estimated as

$$T_{\text{trans}} \sim \delta \cdot 6\pi\mu r \cdot v, \quad (2)$$

with dynamic viscosity $\mu = 1.0 \times 10^{-3}$ Pa · s. Substituting values yields

$$T_{\text{trans}} \sim 0.0381 \times 10^{-3} \cdot 6\pi \cdot 1.0 \times 10^{-3} \cdot 0.381 \times 10^{-3} \cdot 0.0115 \approx 3.14 \times 10^{-13} \text{ Nm.} \quad (3)$$

In comparison, the applied magnetic torque is given by

$$T_{\text{mag}} = mB, \quad (4)$$

where $B = 1.6$ mT is the magnetic field strength, and $m \sim 10^{-8}$ A · m² is the typical magnetic moment of the particle. This gives

$$T_{\text{mag}} \sim 10^{-8} \cdot 1.6 \times 10^{-3} = 1.6 \times 10^{-11} \text{ Nm}. \quad (5)$$

Therefore, the ratio is

$$\frac{T_{\text{trans}}}{T_{\text{mag}}} \approx \frac{3.14 \times 10^{-13}}{1.6 \times 10^{-11}} \approx 0.0196 \approx 2\%, \quad (6)$$

indicating that even under conservative assumptions, the translation-rotation coupling induced torque is less than 2% of the magnetic torque. Of course specific estimations may change based on different parameter settings of the magnetic field, particle properties and the background turbulence.

Note that in our system the phase lag between the particle rotation and the applied magnetic torque is a key observable for identifying stochastic resonance. If translational motion were to induce non-negligible torques: (i) The effective torque acting on the particle would become a combination of magnetic torque, viscous drag, and translationally-induced torque. (ii) This additional component could either amplify or suppress the rotational response, depending on its sign and correlation with the magnetic forcing. (iii) As a result, the phase lag might represent a small shift, and the location of the stochastic resonance peak (i.e., the optimal magnetic frequency where lag is maximized) could move or broaden.

However, because these translation-rotation coupling induced torques are uncorrelated with the externally applied rotational magnetic field and are typically noisy, their effect is expected to increase rotational fluctuations but not generate coherent resonance behavior. Therefore, their presence may reduce the sharpness of the stochastic resonance peak, but not eliminate it.

Crucially, the emergence of stochastic resonance in our system arises from the interplay between external periodic forcing and intrinsic rotational noise. If translation-rotation coupling were to increase the noise level slightly, it would shift the resonance condition, but the resonance mechanism itself remains intact. This means that our key conclusion – that maximal phase lag emerges when magnetic forcing balances turbulent fluctuations – remains qualitatively valid.

To address this concern more explicitly, we have added a paragraph discussing these potential effects in the revised manuscript.

The modifications in the SI of the paper are listed below:

In the model of particle angular dynamics, we neglect rotation–translation coupling by assuming that the particle center of mass and hydrodynamic center are approximately aligned. This assumption is justified by the nearly spherical shape, uniform mass distribution, and small size of the particles (compared to the Kolmogorov scale), which collectively minimize translation-induced hydrodynamic torque. Consequently, the hydrodynamic torque induced by translational motion is expected to be negligible. Even if weak coupling exists, its effect can be interpreted as an additional source of rotational noise that is uncorrelated with the

external magnetic field. This may slightly modify the amplitude of angular fluctuations or slightly shift the resonance frequency, but it does not alter the underlying mechanism of stochastic resonance. Since the emergence of stochastic resonance in our system fundamentally arises from the balance between external magnetic forcing and intrinsic rotational noise, the presence of small translation–rotation coupling does not qualitatively affect our main conclusions.

Comment 8: *Overdamped Approximation*

The overdamped limit is central to the analysis. The authors should discuss its validity more thoroughly:

- Under what conditions (e.g. magnetic field strength, frequency, particle size/moment inertia/magnetization) is the approximation justified?

- Have they evaluated rotational acceleration statistics to confirm the absence of inertial effects beyond first moments?

Response: We now provide a more detailed justification for the conditions under which the rotational inertia of the particle can be neglected, both from dimensional analysis and experimental evidence.

We first discuss the conditions that justify the overdamped approximation for particle rotational dynamics.

The validity of the overdamped approximation for rotational dynamics depends on the relative importance between inertial effects and viscous (drag) effects. Specifically, it is justified when the characteristic time scale for rotational inertia is significantly smaller than the characteristic time scale of the driving forces or the fluid rotational motion. For a light spherical particle, the rotational response time (τ_{rot}) quantifies how quickly a particle rotation adapts to changes in the surrounding fluid or applied torques. As previously derived in the reply to comment 5, τ_{rot} can be estimated as $\tau_{rot} = \frac{\rho_p r^2}{15\mu}$. Based on the experimental parameters, we have $\tau_{rot} = 2.01$ ms, which is smaller than the characteristic time scale of the fluid vorticity, $\tau_\eta = 7.56$ ms. This indicates that the particle can respond significantly faster than the characteristic changes in the fluid rotational fields. This rapid response allows us to neglect the rotational inertial term (angular acceleration) in the rotational equation of motion, simplifying it to a balance between applied torques (magnetic) and viscous drag.

Now we discuss the specific conditions of several related parameters. With respect to particle size (r) and moment of inertia (I), as shown by the formula, τ_{rot} is directly proportional to $\rho_p r^2$ (which is proportional to the moment of inertia I). Smaller particles with lower inertia will naturally exhibit a shorter τ_{rot} , further justifying the overdamped limit. Our chosen particle size lies within a regime where this condition holds relative to the turbulent time scales. While the magnetic torque is a significant driving mechanism, the overdamped approximation primarily concerns the particle response time to any applied torque (magnetic or hydrodynamic). As long as the frequency of the magnetic field and the resulting particle rotation are slower than or comparable to $1/\tau_{rot}$, the particle rotational motion will remain in the overdamped regime. If the magnetic field were to oscillate at frequencies significantly higher than $1/\tau_{rot}$, then inertial effects would become relevant as the particle would struggle to instantaneously follow the rapidly changing magnetic field. Our experimental frequencies are chosen to ensure the particle can follow the driving field within the overdamped regime.

The observed good agreement between our numerical simulations (which is built on the overdamped assumption, i.e., neglecting the angular acceleration term $I\dot{\omega}_p$) and the experimental measurements of

particle angular velocity provides strong direct evidence. This suggests that rotational acceleration is indeed negligible on average, and its fluctuating component does not dominate the dynamics to the extent that it invalidates the overdamped model. The relatively small ratio of τ_{rot}/τ_η further supports this by indicating that angular velocity fluctuations occur on time scales where the particle can rapidly adjust, minimizing the build-up of significant inertial terms.

In summary, the combined analytical estimation of τ_{rot} relative to the flow characteristic time scales, the physical parameter settings of our system, and the strong empirical validation from experimental comparison collectively justify the use of the overdamped approximation for particle rotation in our study.

We have clarified this in the revised manuscript.

The modifications in the SI are listed below which is combined with that in the reply to comment 5:

To justify the validity of the overdamped model, we carefully assessed inertial effects. For a spherical particle, the rotational response time can be estimated as $\tau_{rot} = \frac{I}{\xi_r}$, where $I = \frac{2}{5}\mathcal{V}_p\rho_p r^2$ is the particle moment of inertia, and $\xi_r = 8\pi\mu r^3$ is the rotational drag coefficient. Substituting these into the equation for τ_{rot} simplifies to $\tau_{rot} = \frac{\rho_p r^2}{15\mu}$. Using our experimental parameters of $r_p = 0.381$ mm, $\rho_p = 208$ kg/m³, $\mu \approx 1.0 \times 10^{-3}$ Pa · s, and $\tau_\eta = 7.56$ ms. The rotational response time is $\tau_{rot} \approx 2.01$ ms. We observe that τ_{rot} is approximately $2.01/7.56 \approx 0.266$ or about one-quarter of τ_η , indicating that the particle rotational dynamics are significantly faster than the characteristic changes in the flow vorticity. Therefore, the particle can respond quickly to the flow vorticity, making the assumption of an overdamped regime a reasonable first approximation. Note that our experimental frequencies are chosen to ensure the particle can follow the driving field within the overdamped regime, i.e., the frequency of the magnetic field ω_H is slower than $1/\tau_{rot} \approx 2000$ Hz, so that the particle rotational motion will remain in the overdamped regime.

Comment 9: *Effect of Von Kármán Flow Properties*

While the Von Kármán setup is a standard turbulence generator, it is not isotropic or homogeneous, particularly due to its central core with non-zero mean helicity. Could this large-scale anisotropy influence the observed symmetry breaking or other phenomena? A discussion on this point would improve the interpretation of results and their generalizability.

Response: The Von Kármán-type flow is indeed not perfectly homogeneous or isotropic but it features an intense turbulent region at its center and is homogeneous in the radial directions, i.e., parallel to the two rotating disks. The axial velocity component typically exhibits stronger intermittency than its radial counterparts. As a consequence, for example, the velocity autocorrelation function decays more rapidly in the axial direction. Though it is true that the large-scale helicity and inhomogeneity may influence global flow organization and intermittency in the axial direction, many studies have shown that a fully developed turbulent flow can be generated in the central region of the container, and the small-scale turbulent statistics approach local isotropy, with velocity gradients and dissipation-dominated dynamics resembling those of homogeneous and isotropic turbulence (Mordant et al., 2004; Volk et al., 2008; Zocchi et al., 1994; Voth et al., 2002; Mordant et al., 2001; La Porta et al., 2001; Volk et al., 2011). This region, measuring approximately $20 \text{ mm} \times 20 \text{ mm} \times 20 \text{ mm}$, serves as the primary domain for measurements.

In our experiments, the particles are of size $r \sim 5\eta$, such that their rotational dynamics are primarily coupled to the dissipative scales of turbulence. These scales are much less sensitive to the large-scale anisotropy or mean flow structures, and are instead governed by nearly isotropic velocity gradients, especially in the chosen measurement domain. The observed stochastic resonance and symmetry-breaking behaviors thus originate from the interplay between particle-scale turbulent fluctuations and the externally applied magnetic field, rather than from the large-scale anisotropy of the Von Kármán-type flow.

We have now added a discussion of these limitations to the Method section in the revised manuscript to clarify the generalizability of our results.

The modifications in the revised SI are listed below:

While the Von Kármán-type turbulent flow is not perfectly homogeneous and isotropic, a fully developed turbulent flow can be generated in the central region of the container with small-scale statistics close to local isotropy (Mordant et al., 2004; Volk et al., 2008; Zocchi et al., 1994; Voth et al., 2002; Mordant et al., 2001; La Porta et al., 2001; Volk et al., 2011). This region, measuring approximately $20 \text{ mm} \times 20 \text{ mm} \times 20 \text{ mm}$, serves as the primary domain for measurements. Since our small particles ($r \sim 5\eta$) primarily interact with dissipative-scale velocity gradients in this central region, the observed stochastic resonance and symmetry breaking are governed by particle-scale turbulence and magnetic forcing rather than large-scale flow anisotropy.

Closing remarks

The authors thank the reviewer once more for the many detailed comments on the paper, which helped to improve the manuscript. We have implemented all the suggestions in the revised paper. We hope that with these changes and additions, the paper can now be given a full recommendation for publication in Nature Communications.

References

- Balachandar, S. and Eaton, J. K. (2010). Turbulent dispersed multiphase flow. Annual review of fluid mechanics, 42(1):111–133.
- Belovs, M. and Čēbers, A. (2006). Dynamic fluctuations of dipolar semiflexible filaments. Physical Review E—Statistical, Nonlinear, and Soft Matter Physics, 73(2):021507.
- Dhar, P., Swayne, C. D., Fischer, T. M., Kline, T., and Sen, A. (2007). Orientations of overdamped magnetic nanorod-gyrosopes. Nano letters, 7(4):1010–1012.
- La Porta, A., Voth, G. A., Crawford, A. M., Alexander, J., and Bodenschatz, E. (2001). Fluid particle accelerations in fully developed turbulence. Nature, 409(6823):1017–1019.
- Labbé, R., Pinton, J.-F., and Fauve, S. (1996). Study of the von kármán flow between coaxial corotating disks. Phys. Fluids, 8(4):914–922.
- Maxey, M. R. and Riley, J. J. (1983). Equation of motion for a small rigid sphere in a nonuniform flow. Physics of Fluids, 26(4):883–889.
- Melle, S., Calderón, O. G., Rubio, M. A., and Fuller, G. G. (2002). Chain rotational dynamics in mr suspensions. International Journal of Modern Physics B, 16(17n18):2293–2299.
- Mordant, N. (1997). Characterization of turbulence in a closed flow. J. Phys. II France, 7:1729–1742.
- Mordant, N., Lévêque, E., and Pinton, J.-F. (2004). Experimental and numerical study of the lagrangian dynamics of high reynolds turbulence. New Journal of Physics, 6(1):116.
- Mordant, N., Metz, P., Michel, O., and Pinton, J.-F. (2001). Measurement of lagrangian velocity in fully developed turbulence. Physical Review Letters, 87(21):214501.
- Osborn, J. A. (1945). Demagnetizing factors of the general ellipsoid. Physical Review, 67(11-12):351.
- Qian, B., Powers, T. R., and Breuer, K. S. (2008). Shape transition and propulsive force of an elastic rod rotating in a viscous fluid. Physical review letters, 100(7):078101.

- Tierno, P., Claret, J., Sagués, F., and Cēbers, A. (2009). Overdamped dynamics of paramagnetic ellipsoids in a precessing magnetic field. Physical Review E, 79(2):021501.
- Toschi, F. and Bodenschatz, E. (2009). Lagrangian properties of particles in turbulence. Annual Review of Fluid Mechanics, 41:375–404.
- Volk, R., Calzavarini, E., Leveque, E., and Pinton, J.-F. (2011). Dynamics of inertial particles in a turbulent von kármán flow. Journal of Fluid Mechanics, 668:223–235.
- Volk, R., Calzavarini, E., Verhille, G., Lohse, D., Mordant, N., Pinton, J.-F., and Toschi, F. (2008). Acceleration of heavy and light particles in turbulence: comparison between experiments and direct numerical simulations. Physica D: Nonlinear Phenomena, 237(14-17):2084–2089.
- Voth, G. A., La Porta, A., Crawford, A. M., Alexander, J., and Bodenschatz, E. (2002). Measurement of particle accelerations in fully developed turbulence. Journal of Fluid Mechanics, 469:121–160.
- Wu, C., Kunnen, R. P. J., Wang, Z., de Wit, X. M., Toschi, F., and Clercx, H. J. H. (2025). Tracking the rotation of light magnetic particles in turbulence. arXiv preprint arXiv:2506.21769.
- Zimmermann, R., Gasteuil, Y., Bourgoïn, M., Volk, R., Pumir, A., Pinton, J.-F., et al. (2011). Tracking the dynamics of translation and absolute orientation of a sphere in a turbulent flow. Review of Scientific Instruments, 82(3).
- Zocchi, G., Tabeling, P., Maurer, J., and Willaime, H. (1994). Measurement of the scaling of the dissipation at high reynolds numbers. Physical Review E, 50(5):3693.

Manuscript: Stochastic resonance of rotating particles in turbulence

Tracking: NCOMMS-25-34227-T

Authors: Ziqi Wang, Xander M. de Wit, Roberto Benzi,
Chunlai Wu, Rudie P. J. Kunnen, Herman J. H. Clercx,
and Federico Toschi

September 5, 2025

Response to Reviewer 2

We would like to thank the reviewer for the thorough reading of our manuscript, for his/her many insightful comments, and suggestions that allowed us to considerably improve our manuscript. We really appreciate these efforts by the reviewer.

Synopsis: *The authors report on the rotational dynamics of a magnetic particle (subjected to a rotating magnetic field) in a turbulent Von Kármán flow (generated by two counter-rotating impellers). They observe three particle-rotation regimes experimentally and numerically: A phase-locked regime synchronized with the field rotation, a back-and-forth particle motion, and a turbulence-dominated one. The main result of the manuscript concerns the numerical evidence of stochastic resonance of the particle angle occurring for intermediate stochastic noise of turbulence.*

To do so, the authors built an elegant experimental system and developed a numerical method to probe the stochastic resonance. Although redundancies and back-and-forth are too frequent, the manuscript is rather well written, and the figures are presented very cleanly. However, I find that the experimental part is too reduced and concerns only the phase-locked regime and the back-and-forth regimes, which are not new in the literature, the latter being not sufficiently quoted. Moreover, I have several serious concerns about the claims made by the authors.

Response: We are very grateful to the reviewer for all the suggestions on how to improve our manuscript. In line with the comments of the reviewer, we have made a sincere and dedicated effort for the revision of the manuscript. In addition, a point-by-point response to all comments/suggestions is provided below. Please note that all modifications made in the revised manuscript and Supplementary Material based on the comments of the reviewer are highlighted in blue. For convenience, here we have also quoted the modified

content in the revised main paper and the revised Supplementary Material and highlighted them in blue. In the following, we answer/clarify all points raised by the reviewer.

Comment 1: *The synchronized regime and the back-and-forth regime are known phenomena occurring without noise, as already observed experimentally for a compass (V. Croquette et al., J. Phys. Lett. 1981, A. Poyé et al., Phys. Scr. 2019) or a rolling particle (A. Kaiser et al., Sci. Adv. 2017) in an alternating magnetic field. On the other hand, the turbulent regime occurs when the effect of the magnetic field is negligible, which, in practice, corresponds to a nonmagnetic particle in a turbulent flow. Thus, only the results related to the intermediate noise level are new, including the stochastic resonance. It should be clearly stated in the manuscript, and these references should at least be discussed. The bibliography also fails to include key articles relevant to the topic of this manuscript (see below).*

Response: We thank the reviewer for this insightful comment and for highlighting relevant literature, which provides an excellent opportunity to more precisely articulate the novelty and scope of our work.

We first clarify the distinctions between our work and the cited studies (Croquette and Poitou, 1981; Poyé et al., 2018; Kaiser et al., 2017). While our work, like the cited papers, explores collective behaviors of magnetic particles and phenomena such as phase locking and oscillation (back and forth), there are fundamental differences in the driving mechanism of the system and experimental conditions:

- **Background fluid setting:** A key aspect of our system, and a major departure from the cited works, is the presence of turbulent flow. And the effect of turbulence on magnetic particles within a rotating magnetic field is an unexplored phenomenon. It is not known a priori whether this effect can be analogously modeled as simple white noise. The mentioned papers, which use quiescent fluid (air or isopropanol), do not include the complex interplay between particles, turbulent flow, and the magnetic torque. This tripartite interaction is central to the novel phenomena observed in our study and is not present in the more simplified systems previously investigated. The stochastic resonance phenomenon in our work is a direct consequence of this complex interplay, not merely a state where the magnetic field effect is negligible. This is in agreement with the comment of the Referee, that the interesting dynamics that is the focus of this paper indeed lies in the intermediate region where both the magnetic field and the turbulence play a central role.
- **Magnetic field configuration:** Our research employs a rotating magnetic field, meaning the magnetic field direction continuously rotates around a fixed axis while its magnitude remains constant. This is a crucial distinction from the cited works, which investigate systems under an oscillating magnetic field with a fixed direction (i.e., the field direction is fixed, but its strength varies periodically with time).
- **Particle volume fraction:** Kaiser et al. (2017) focuses on magnetic particle solutions in high volume fraction regimes, where hydrodynamic interactions between particles are involved. In contrast, our

current research operates in a dilute regime, meaning the particle concentration is significantly lower, which fundamentally alters the nature of particle-particle interactions and the emergence of collective phenomena.

- **Particle properties:** Croquette and Poitou (1981); Poyé et al. (2018) use a compass as their “particle” in their experiment. Kaiser et al. (2017) uses micrometer-scale heavy (w.r.t. water) particles. While in our work, we use light, millimeter scale particles, which have crucially different Lagrangian dynamics.

We agree with the observation that, in ideal quiescent fluid systems, the synchronized and back-and-forth regimes have indeed been previously observed, as demonstrated by the experimental work on compasses (Croquette and Poitou, 1981; Poyé et al., 2018) or rolling particles (Kaiser et al., 2017) in alternating magnetic fields. We also agree that the “turbulent regime,” where the magnetic field influence becomes negligible, is analogous to the behavior of a non-magnetic particle passively rotating in a turbulent flow.

However, the core novelty of our work lies not in the isolated existence of these regimes, but in the systematic, quantitative investigation of their interactions, transitions, and statistical properties within the complex, stochastic environment of fully developed turbulence. Our work explores an uncharted territory: the complex interplay between deterministic external magnetic forcing and the chaotic, stochastic hydrodynamic torques arising from small-scale turbulent vorticity. It is exactly this complex interplay results in the stochastic resonance phenomena of magnetic particles which is the first time to be explored in turbulent flows, representing a fundamentally new area of exploration.

Specifically, our contributions introduce several novel aspects:

- **Exploring a new phase space under turbulent forcing:** To our knowledge, our study provides the first comprehensive exploration of the rotational dynamics of magnetically driven light particles across a broad range of magnetic field strengths and frequencies within a turbulent flow environment. This reveals a significantly richer and more complex phase space of rotational states, distinct from those observed in quiescent or laminar fluids. We characterize how the stochastic nature of turbulence fundamentally alters and transitions between these rotational behaviors.
- **Turbulence as an active noise source for stochastic resonance:** Our most significant novel finding is the first observation and detailed characterization of stochastic resonance (SR) in the angular velocity of light particles immersed in turbulence. We observed that SR emerges at the transition from the phase-lock regime to the turbulent dynamics regime, which further highlights the crucial interplay between magnetic and hydrodynamic torques. Turbulent fluctuations, remarkably acting as an effective noise, are shown to enhance the particle rotational response to the external periodic magnetic forcing. This counterintuitive phenomenon, where an optimal level of turbulent

noise amplifies the system response, opens new avenues for understanding and exploiting turbulence-particle interactions. While SR has been widely studied in various systems (Benzi et al., 1981; Gammaitoni et al., 1998), its manifestation and underlying mechanisms in the rotational dynamics of particles directly driven by turbulent fluctuations have remained largely unexplored.

- **Discovery of a novel symmetry-breaking mechanism:** We reveal a unique symmetry-breaking mechanism where an oscillating magnetic field with zero-mean angular velocity can induce a net particle rotation in a turbulent flow with zero-mean vorticity. This phenomenon, where turbulent fluctuations aid the particle in “surfing” the magnetic field, represents a qualitatively new dynamic state not observed in low-noise or deterministic settings.
- **Novel application for turbulence probing:** Leveraging these discovered phenomena, particularly SR and the symmetry-breaking mechanism, our work proposes and lays the groundwork for novel techniques for flexibly manipulating particle dynamics in complex flows and developing a magnetic resonance-based approach for measuring turbulent vorticity even under optically inaccessible conditions. These applied aspects directly stem from the unique dynamics uncovered by introducing turbulence as an active component.

To address the reviewer’s valuable feedback, we revised the manuscript to:

- Explicitly state that while synchronized and back-and-forth regimes exist in many physical systems, such as magnetic particles driven by rotating magnetic field in quiescent fluid and magnetic colloids in alternating magnetic field, our novelty arises from (i) the complex interplay between the turbulence induced hydrodynamic torque and the rotating magnetic field induced magnetic torque, and (ii) the transitions (evidenced as a stochastic resonance phenomenon) observed within a turbulent environment. We mainly focus on the role of stochasticity of turbulence and the emergence of stochastic resonance.
- Incorporate and discuss the suggested foundational references (Croquette and Poitou, 1981; Poyé et al., 2018; Kaiser et al., 2017) to properly contextualize our findings, highlighting how our work extends beyond these studies by introducing and exploring the intricate role of turbulence.
- Thoroughly review and update the bibliography to ensure all key articles relevant to the topic of magnetically driven particles, turbulent particle dynamics, and stochastic resonance are appropriately cited and discussed.

We believe these revisions will significantly enhance the clarity of our contributions and properly position our work within the existing literature.

The modifications in the main paper are listed below:

In the Introduction part, we have added:

Here, we explore this question by investigating the rotational dynamics of magnetic particles in turbulence under a rotating magnetic field. While deterministic synchronized and oscillation (back-and-forth) rotational regimes have been observed in zero-noise systems (e.g., compasses or rolling particles in quiescent fluids Croquette and Poitou (1981); Poyé et al. (2018); Kaiser et al. (2017)), our work extends this understanding into the complex environment of fully developed turbulence. We systematically explore the complex interplay between deterministic external magnetic forcing and turbulent hydrodynamic torques, revealing a significantly richer particle phase space of rotational dynamics. We show that the effect of turbulence (vorticity) acts as an effective noise (Supplementary Information) and reveal the first signature of the emergence of stochastic resonance (SR) ...

In the Supplementary Information, we have added:

Previous studies have indeed explored the rotational dynamics of magnetically driven objects in quiescent environments (Croquette and Poitou, 1981; Poyé et al., 2018; Kaiser et al., 2017; Yan et al., 2012; Cīmurs et al., 2019; Cēbers and Ozols, 2006; Ērglis et al., 2007; Cīmurs and Cēbers, 2013; Morozov et al., 2017), demonstrating phenomena such as phase locking and oscillatory motion. However, a fundamental distinction of our work lies in investigating these dynamics within the stochastic and chaotic environment of fully developed turbulence. This critical inclusion reveals a new class of phenomena, particularly the emergence of stochastic resonance where turbulent fluctuations play an active role in enhancing the particle's response to an external periodic magnetic field. Furthermore, we uncover a novel symmetry-breaking mechanism for inducing net particle rotation in zero-mean turbulent vorticity.

Comment 2: *The authors want to address an important question: Understanding the coupling between a particle’s rotation and the underlying vorticity field in a turbulent flow. The authors state that magnetic particles can control the vorticity of the flow. In turbulent flows, energy is generally injected at large scales (e.g., by an impeller) and cascades towards small scales. In this system studied by the authors, the energy contained at the particle size is therefore much smaller than the energy-containing range, and the particle rotation will surely influence only slightly the dynamics of the flow and its vorticity. The authors could comment on that point.*

Such claims could have been supported by experimental measurements in the flow velocity field, but the manuscript appears to make limited use of the experimental facility. Most of the results are numerical simulations, in particular, the main result of stochastic resonance in turbulence, which I believe could have also been evidenced experimentally. Consequently, the significance of this work to the field is questionable, mainly due to the lack of experimental data to compare with the numerical resolution of a single particle equation. Reading the abstract, I would have expected more experimental results (only Figs. 1 and 2 are experimental), and the main result of the manuscript is numerical. Moreover, the authors use optical methods to track the particle and measure the flow vorticity. I believe measurements with PTV or PIV would also give very high-quality vorticity data and probably be less intrusive to the flow than using particles with a size above the Kolmogorov length scale.

Response: Following the reviewer’s suggestion, we have performed additional experiments that provide clearer evidence of stochastic resonance, from the experimental point of view. These new results are now included in the revised manuscript, as shown by Fig. 4(a) and (c) in the main text. For clarity, we also sum up the new results here as Fig. R1(a) (in the reply to the reviewer’s comment 9), with the diamond symbols indicating the newly added experimental data. More discussion about these additional experiments can be found in the reply to the reviewer’s comment 9.

Next, we will explain each of the reviewer’s concerns in turn.

First, the strong tendency to universality (homogeneous and isotropic state) at small scales in turbulence severely limits our capability to modulate turbulence by acting only on large-scale degrees of freedom (Arneodo et al., 1996; Frisch, 1995). Indeed, traditional approaches such as boundary structuring or actuation for the control of laminar flows in microfluidic devices (Stroock et al., 2002) are generally ineffective at penetrating into the bulk turbulent flow. In contrast, examples such as polymer additives or bubbles show that when disturbances are able to interact with turbulence throughout the full volume, even at very small volume fractions, they can induce dramatic changes in the global flow properties (e.g. drag reduction) (Van Gils et al., 2013; Toms, 1949; van den Berg et al., 2007). Some research efforts have already indicated the possibility to numerically or experimentally force turbulence at the small scales and change the statistical properties and dissipation in turbulent flows (Buzzicotti et al., 2020; Falcon et al., 2017). In particular, we have recently proved numerically that forced light particles can strongly reduce the intermittency of

the turbulent flow (Freitas et al., 2025). These examples highlight that it is precisely the physics at the smallest scales that can give rise to non-linear, system-level responses.

In this context, the ultimate objective of our broader-scale project is to explore whether turbulence can be influenced from within, at its smallest scales, through externally actuated particles. That is why our particles are chosen to be small (of a size comparable to the Kolmogorov scale), light (preferential concentration in the vortex (Calzavarini et al., 2008; Mathai et al., 2020; Wang et al., 2024)), and magnetic (external controllable angular dynamics) to interact with helicity and small-scale structures, thereby “touching turbulence at its core.” To move towards this ambitious objective, the present work should be seen as a proof of concept and focuses on the essential prerequisite: gaining a quantitative understanding of the coupled dynamics of magnetic particles under external forcing and turbulent fluctuations. We demonstrate that even at the particle scale, controlled interactions can lead to detectable dynamical effects such as stochastic resonance. While the absolute energy content at the particle size is indeed smaller, our long-term goal is to exploit ensembles of smart particles to collectively shape turbulence from within.

Second, concerning the balance of experimental and numerical methods: we acknowledge that the current manuscript relies more on numerical simulations. This choice was made because numerical simulations provided us with a controlled environment where we can isolate the physics of particle–flow coupling before moving to more complex and costly experiments. Nevertheless, the experiments already play a crucial role in validating the essential mechanisms: (1) they confirm the statistical angular dynamics across different phase regimes (i.e., phase-locked, turbulent dynamics, and back-and-forth), as shown by the PDFs in Fig. 2 in the main text; (2) they provide experimental evidence of stochastic resonance, which has been investigated recently and the newly obtained results have been included in the revised manuscript (Fig. 3(m), Fig. 4(a) and (c) in the main text). We believe that with this revision, our results now strongly demonstrate both the feasibility of our experimental approach and the physical mechanisms underlying the complex particle-magnetic field-flow interaction.

Third, we would like to emphasize that the goal of the present work is not to present a ready-to-use platform to perform global vorticity measurements of the flow field, but our work can be used as a first step to inspire such future work. Our work provides an accurate understanding of the complex interplay between hydrodynamic and magnetic torques. Our experimental approach was designed to directly probe the rotational dynamics of magnetically actuated particles. We agree that more extensive experimental investigation, ideally including velocity-field measurements (PIV/PTV), would provide complementary information and very high-quality vorticity data. Such measurements, however, are beyond the scope of the present study and form part of our ongoing and future work.

In summary, we stress that the present work should be understood as a first, but substantial, step towards the broader goal of “shaping turbulence with smart particles”. The current combination of experiments and simulations demonstrates feasibility and highlights new physics (such as stochastic resonance), while more extensive experimental investigation will be the focus of our future studies.

Comment 3: *The main message of the manuscript is that vorticity fluctuation effects on a magnetic particle can be seen as “noise” without the need to solve complex hydrodynamics equations. However, I do not see any results showing the impact of the particle’s rotation on the flow via hydrodynamic interactions, which is stated in the introduction as a mechanism to control turbulence.*

Response: First, we clarify the scope of the present work. The ultimate goal of our research is indeed to explore the possibility of controlling turbulence through interactions with forced particles. However, to reach this ultimate objective, a crucial first step is to understand the dynamics of such particles under the influence of turbulence and external forcing.

The current work focuses precisely on this first foundational step in that direction: we investigate how magnetic particles respond to turbulent vorticity when their rotational degrees of freedom are subject to external magnetic forcing. We demonstrate that, by tuning the relative strengths and frequencies of the external forcing, it is possible to induce stochastic resonance – thereby maximizing the particle rotational response. This demonstration is crucial, as it establishes that such magnetic particles can be made to resonate with turbulent fluctuations in a statistically predictable and externally controllable manner, without the need to resolve the full hydrodynamics in every instance.

Thus, the current study does not yet address how the particle rotation alters the surrounding flow, but rather lays the groundwork by showing that the rotational dynamics of the particles can be robustly and predictably controlled. This is a necessary prerequisite for any future attempt at particle-based turbulence control.

Although the control of turbulence via particle-induced feedback is a highly nontrivial task, there is increasing evidence suggesting that such a strategy is feasible, especially if one targets the small scales of turbulence where coherent structures, such as vortex filaments, play a critical role in determining intermittency, energy cascade, and dissipation statistics. Controlling these small-scale features is arguably the most efficient way to influence global turbulence properties. For example, Buzzicotti et al. (2020) shows that the presence of a smart small-scale drag (by introducing a nonlinear extra viscosity acting preferentially on high vorticity regions) can strongly reduce intermittency and non-Gaussian fluctuations of turbulence. And we have recently demonstrated the next step, showing from numerics that this can be practically achieved using forced light particles (Freitas et al., 2025).

In our work, magnetic particles provide a promising new pathway for influencing turbulence. Because they can be actively actuated by external magnetic fields, their rotation can be tuned to resonate with turbulent fluctuations, thereby creating a mechanism for localized and targeted energy injection or extraction. This feature is in stark contrast to passive or inertial particles, whose response is dictated by the flow alone. The demonstrated stochastic resonance behavior in our study thus opens a promising avenue for future flow control applications.

We fully agree with the reviewer of the importance for evaluating the back-reaction of particle motion on the turbulent flow. Indeed, this is a key focus of the next phases of our ongoing research. Our next steps include: (i) Measuring the background Eulerian flow field of turbulence in the presence and absence of magnetic particles under externally applied rotating magnetic field; (ii) Quantifying the changes in flow statistics such as velocity structure functions, acceleration probability density functions, and energy spectra; (iii) Investigating how controlled particle rotation influences the generation, persistence, or suppression of small-scale vortex filaments. These analyses will allow us to rigorously determine whether and how magnetic particles can exert feedback on turbulence, moving us closer to the long-term goal of flow manipulation via smart particles. However, at the root of all such future directions, is the fundamental knowledge of the dynamics of magnetic particles under the influences of turbulence and applied magnetic forcing, for which this work lays the foundation.

We have added a paragraph in the Discussion section to clarify the scope of the current manuscript and to explicitly point to the follow-up work.

The modifications in the main paper are listed below:

While the present work focuses on understanding particle angular dynamics, it lays essential groundwork for subsequent investigations into how these actively tunable particles can shape turbulent structures, ~~Our findings also~~ opening possibilities for flexible manipulation of turbulence, where propeller-like particles could be tuned to navigate specific flow structures, inject energy into turbulence, and influence flow properties. More broadly, our study provides fundamental mechanisms that externally tunable rotational dynamics could be leveraged to design active materials Scholz et al. (2018); Yang and Zhang (2021); Jiang et al. (2023). If organized coherently, their collective behavior may also exhibit hydrodynamic properties analogous to systems with odd viscosity, which may be promising experimental realizations of chiral fluids Chen et al. (2025); Fruchart et al. (2023).

Comment 4: *There is a claim in the abstract, introduction, and conclusion that using magnetic particles to probe turbulent fluctuations can circumvent optical techniques. However, the experimental results in the manuscript solely rely on optical measurements. Can the author clarify what they have in mind?*

Response: We appreciate the reviewer’s careful reading and this opportunity to clarify our intent.

Indeed, the experiments presented in the current manuscript rely on optical tracking to validate the feasibility of using magnetic particles as probes for turbulent vorticity. However, the core idea we propose goes beyond these specific experiments and aims to outline a potential measurement strategy that could be used in *optically inaccessible conditions* for future applications.

Specifically, once the relationship between the angular response of magnetic particles and the local turbulent vorticity is well-characterized (as we aim to establish with our current optically-based experiments), it becomes possible to provide a protocol where the magnetic signature emitted by the particles themselves (e.g., via magnetic dipole field measurements) can be detected by magnetic sensors placed outside the flow. In such a setup, the rotation dynamics of the particles can be reconstructed from the modulated magnetic field they emit, making optical access unnecessary.

This concept has strong analogies with magnetometry techniques used in biomedical applications, where the dynamics of magnetic beads are detected through remote magnetic sensors. Our long-term vision is to transfer this idea into turbulence diagnostics: to design a “magnetic vorticity microscope” that functions even in opaque or confined environments (e.g., opaque suspensions, industrial flows, liquid metals).

To avoid confusion, we have now revised the relevant parts of the abstract, introduction, and conclusion to make clear that while the *current demonstration relies on optical methods*, the magnetically-driven probing strategy itself is not inherently optical, and could be adapted for non-optical implementation in future work.

The modifications in the main paper are listed below:

Different particle properties influence their response, and when combined with the stochastic resonance mechanism, they enable a novel magnetic resonance-based approach to measure turbulent vorticity. This method leverages activated magnetic particles as probes, effectively functioning as a turbulent vorticity “microscope” that operates in various conditions. In the present work, we validate this concept using optical measurements of particle rotation. However, in principle, once the angular response is well-characterized, the same protocol could be applied even in optically inaccessible environments by measuring the magnetic fields emitted by the magnetic spinners directly. In such scenarios, the activated magnetic particles would function as remotely readable vorticity probes. ~~When the magnetic field emitted by the magnetic spinners can be measured directly, it can even be used in optically inaccessible conditions.~~ The protocol is straightforward ...

Comment 5: *How do the authors measure the Kolmogorov timescale in their experimental setup, presented in Fig. 2? What is the experimental Stokes' number?*

Response: We firstly measured the dissipation rate ε that is determined from the driving torque on the two impellers as $\varepsilon = T\Omega/(\rho V_w)$ (Mordant, 1997; Labbé et al., 1996), where T is the sum of the driving torques from the impellers at the top and bottom of the water tank and $V_w = 2.8 \times 10^{-3} \text{ m}^3$ is the volume of the water tank. The driving torques are measured using two strain gauge torque meters mounted on the motors. In our experiments, we investigated two turbulence intensities: (1) the weak-turbulence condition, for which the results are shown in Fig. 2(b-d) of the main text; and (2) the strong-turbulence condition, for which the results are presented in Supplementary Fig. 1(b). For the weak-turbulence case, the impellers rotate at 0.83 Hz. The corresponding mean energy dissipation rate is $\varepsilon = 0.037 \pm 0.002 \text{ m}^2/\text{s}^3$, which yields a Taylor-scale Reynolds number of $Re_\lambda = 398$, a Kolmogorov length scale of $\eta = 0.072 \text{ mm}$, and a Kolmogorov time scale of $\tau_\eta = 5.20 \text{ ms}$. For the strong-turbulence case, the impellers rotate at 1.21 Hz. The corresponding mean energy dissipation rate is $\varepsilon = 0.075 \pm 0.001 \text{ m}^2/\text{s}^3$, which yields a Taylor-scale Reynolds number of $Re_\lambda = 447$, a Kolmogorov length scale of $\eta = 0.061 \text{ mm}$, and a Kolmogorov time scale of $\tau_\eta = 3.66 \text{ ms}$.

We have now included these information in the revised SI.

The modifications in the SI are listed below:

The measured dissipation rate ε is determined from the driving torque on the two impellers as $\varepsilon = T\Omega/(\rho V_w)$ (Mordant, 1997; Labbé et al., 1996), where T is the sum of the driving torques from the impellers at the top and bottom of the water tank and $V_w = 2.8 \times 10^{-3} \text{ m}^3$ is the volume of the water tank. The driving torques are measured using two strain gauge torque meters mounted on the motors. In our experiments, we investigated two turbulence intensities: (1) the weak-turbulence condition, for which the results are shown in Fig. 2(b-d) of the main text; and (2) the strong-turbulence condition, for which the results are presented in Supplementary Fig. 1(b). For the weak-turbulence case, the impellers rotate at 0.83 Hz. The corresponding mean energy dissipation rate is $\varepsilon = 0.037 \pm 0.002 \text{ m}^2/\text{s}^3$, which yields a Taylor-scale Reynolds number of $Re_\lambda = 398$, a Kolmogorov length scale of $\eta = 0.072 \text{ mm}$, and a Kolmogorov time scale of $\tau_\eta = 5.20 \text{ ms}$. For the strong-turbulence case, the impellers rotate at 1.21 Hz. The corresponding mean energy dissipation rate is $\varepsilon = 0.075 \pm 0.001 \text{ m}^2/\text{s}^3$, which yields a Taylor-scale Reynolds number of $Re_\lambda = 447$, a Kolmogorov length scale of $\eta = 0.061 \text{ mm}$, and a Kolmogorov time scale of $\tau_\eta = 3.66 \text{ ms}$.

Comment 6: *The title is misleading and should be explicit, something like “Stochastic resonance dynamics of rotational magnetic particles in turbulence”.*

Response: We agree with the reviewer. To avoid confusion and misleading, we have changed the title to “Stochastic resonance of rotating particles in turbulence”.

Comment 7: *What is the Reynolds number? It should be clearly stated, as is standard in any turbulence study. Depending on its value, the rotational drag coefficient could be laminar (as used in the manuscript) or turbulent. Thus, it could change the conclusion of the manuscript. It should be relatively easy numerically to take this point into account.*

Response: Our experiments are conducted in a Von Kármán-type turbulent flow, characterized by a Taylor-scale Reynolds number of up to $Re_\lambda = 447$, ensuring fully developed homogeneous and isotropic turbulence in the measurement volume. It is noteworthy that the particle rotational dynamics is dominated by the turbulent dynamics at the small scales, i.e. the scales of vortex filaments. It is widely conjectured, motivated by numerical and experimental observations, that the small scales of turbulence are universal Ghira et al. (2022); Kang et al. (2007); Tanahashi et al. (2001); da Silva et al. (2011); Ganapathisubramani et al. (2008). Therefore, we can make the convincing argument that the strength of the large scale forcing, i.e., different values of the Reynolds number, is expected to pose negligible effects on the particle rotational dynamics provided the turbulence is fully developed.

Regarding the rotational dynamics of the particle, the relevant Reynolds number for determining its validity of the particle rotational drag coefficient is the particle rotational Reynolds number, which is defined as $Re_p = \rho_f r^2 |\omega_p - \omega_\eta| / \mu$, where ω_p is the angular velocity of the particle and ω_η is the Kolmogorov vorticity scale. In our experimental system, the maximum instantaneous relative angular velocity between the particle and the fluid is typically found to be $\mathcal{O}(1)$ rad/s. Given our particle radius $r = 0.35$ mm, fluid density $\rho_f \approx 1000$ kg/m³, and viscosity $\mu \approx 10^{-3}$ Pa · s, the rotational Reynolds number for a rotating particle is estimated as $\mathcal{O}(10^{-1}) < \mathcal{O}(1)$. This indicates that the particle motion relative to the local fluid environment and its own rotation occur primarily within the Stokes regime. Therefore, the use of the laminar (Stokes) rotational drag coefficient is fully justified. While the background flow is turbulent, the small particle size and its relative motion ensure that the local viscous forces dominate over inertial effects at the particle scale. Regarding the suggestion to numerically account for non-laminar drag, given that our particle Reynolds numbers are consistently low, introducing more complex, non-linear drag laws (e.g., those accounting for inertial effects at higher Re_p) would lead to negligible improvements in accuracy for the particle dynamics, but significantly increase the model complexity. Our current numerical model provides an accurate leading-order description of the particle-fluid interaction in this low Re_p regime.

We have clarified the relevant Reynolds numbers in the revised manuscript to ensure the validity of our drag coefficient assumption is explicitly stated. Our conclusions, including the observed stochastic resonance and symmetry-breaking mechanism, are robust as they fundamentally depend on the interplay of time scales related to the rotating magnetic field and the turbulent vorticity, which are well-captured by our established model.

The modifications in the revised main paper and SI are listed below:

In the main paper, we have added: *Under these conditions, a fully developed turbulent flow of*

$Re_\lambda = 398$ is generated in the central region of the container (Mordant et al., 2004; Volk et al., 2008; Zocchi et al., 1994; Voth et al., 2002; Mordant et al., 2001; La Porta et al., 2001; Volk et al., 2011).

In the SI, we have added more clarification: *Our experiments are conducted in a Von Kármán-type turbulent flow, characterized by a Taylor-scale Reynolds number of up to $Re_\lambda = 447$, ensuring fully developed homogeneous and isotropic turbulence in the measurement volume. It is noteworthy that the particle rotational dynamics is dominated by the turbulent dynamics at the small scales, i.e. the scales of vortex filaments. It is widely conjectured, motivated by numerical and experimental observations, that the small scales of turbulence are universal (Ghira et al., 2022; Kang et al., 2007; Tanahashi et al., 2001; da Silva et al., 2011; Ganapathisubramani et al., 2008). Therefore, it is expected that, provided the turbulence is fully developed, the strength of the large scale forcing, i.e., different values of the Reynolds number, poses negligible effects on the particle rotational dynamics. While the background flow is turbulent, the particle's rotational dynamics are primarily governed by interactions with small-scale eddies. The particle rotational Reynolds number, $Re_p = \rho_f r^2 |\omega_p - \omega_\eta|$, is estimated to be $\mathcal{O}(10^{-1})$ (below 1) based on typical maximum relative angular velocities, justifying the use of the laminar (Stokes) rotational drag coefficient, ξ_r in out theoretical modelling. This ensures that viscous forces dominate inertial effects at the particle scale, providing an accurate leading-order description of particle-fluid interaction.*

Comment 8: *Why experimental PDFs are skewed in Figs. 2b-d, whereas the signals in Fig. 1g and Fig.1i look symmetric?*

Response: The key distinction lies in the different flow environments represented in these figures. Figures 1g and 1i correspond to the time series of the angular velocity of magnetic particles in a quiescent (non-turbulent) fluid, subject only to an externally imposed rotating magnetic field. In such a controlled environment:

- In Fig. 1g, the particle is in a phase-locked regime, where it continuously rotates in synchrony with the external field. In this regime, the particle maintains a constant angular velocity equal to that of the rotating field. Theoretically, its angular velocity distribution should be a delta function centered at the imposed angular velocity.

- In Fig. 1i, the particle is in a back-and-forth regime, where it undergoes periodic acceleration and deceleration as it tries to follow the rotating field but repeatedly falls behind and catches up. While the PDF of its angular velocity still peaks near the imposed angular velocity, it also includes a range of values due to this dynamic behavior, leading to a broader and slightly skewed distribution.

In contrast, the PDFs shown in Figs. 2b–d represent the angular velocity statistics of magnetic particles in a homogeneous and isotropic turbulent flow. In this environment, the rotation of the particles is influenced not only by the external magnetic field but also by random torques from the turbulent fluid motion. These turbulent perturbations result in intermittent events where: The particle temporarily keeps pace with the external field (phase-locked-like), or lags behind or overshoots (back-and-forth-like).

The resulting angular velocity PDFs thus reflect a mixture of these dynamical regimes, with a prominent peak near the external field's rotation rate, but with a skewed tail due to turbulent fluctuations. The asymmetry arises from this intermittent competition between deterministic magnetic forcing and random turbulent torques.

Comment 9: *It would have been interesting to include more experimental points in Fig. 3 (beyond the three already provided) to perform a reasonable comparison with simulations.*

Response: We thank the reviewer for the suggestion.

We have conducted additional experiments beyond the original three data points, and the new results have been incorporated into the phase diagram in Fig. 4(a) of the main text.

Furthermore, we have included resonance curves in Fig. 4(c) in the main text to provide direct experimental evidence of stochastic resonance. For convenience, the corresponding figures including the new experimental results are also provided here, as shown in Fig. R1.

The diamonds denote the experimental results at $\omega_H/\omega_a = 0.12 \pm 0.01$ (corresponding to the diamonds marked in the phase diagram in Fig. R1a and b). In the resonance curves plot, as shown in Fig. R1c, the circles represent the simulation results at a comparable value of ω_H/ω_a for direct comparison. We chose this value of ω_H/ω_a , because for smaller ω_H/ω_a , the resonance becomes more pronounced. The experimental and simulation results agree reasonably well; at the resonance peak, the experimental curve appears flatter, which can be attributed to polydispersity in particle shape and magnetic properties in experiments, which somewhat washes out the peak by convoluting the results in a small region around the mean ω_η/ω_a , whereas the simulations assume idealized spherical particles with the same magnetic characteristics for each particle.

While time and experimental constraints limit the number of additional points we could obtain, we believe these new data convincingly show the experimental evidence of stochastic resonance, which is the central result in our work, and allow for meaningful comparison with simulations.

Figure R1: **a**, Phase diagram of particle rotation regime. Symbols (square, star, and circle): representative cases from the three distinct regimes shown in earlier panels. Triangle markers: experimental settings from Fig. 2b-d in the main text. The diamond symbols denote the experimental results at $\omega_H/\omega_a = 0.12 \pm 0.01$, providing evidence of stochastic resonance. **b**, Phase diagram of stochastic resonance. The diamond symbols denote the experimental results at $\omega_H/\omega_a = 0.12 \pm 0.01$, providing evidence of stochastic resonance, as shown in c. **c**, Experimental evidence of stochastic resonance (diamonds) at $\omega_H/\omega_a = 0.12 \pm 0.01$. For comparison, simulation results at a similar value of ω_H/ω_a (circles) are also shown. The error bars correspond to the standard deviation calculated from 20 independent measurements at the same parameter value.

Comment 10: *The typical system size and magnetic field strength should appear in the main text (instead of the Methods Section). Generally speaking, the frequent back-and-forth between the main text, Methods section, and Supplemental Information disrupts the reading flow. Some redundancies also appear in these different parts.*

Response: We have added the typical system size and magnetic field strength in the revised main text, and have avoided the redundancies across different parts to improve the readability and flow of the main text. Due to the main text length considerations, we maintain a concise and focused narrative in the main text without disrupting the reading flow, reserving highly detailed experimental protocols and extensive background information for the Methods section and Supplemental Information.

The modifications in the revised main paper are listed below:

The experimental setup consists of a turbulence generator, a magnetic field generator, and magnetic particles. A Von Kármán-type turbulent flow is generated in an octagonal water-filled container with an internal diameter of $2R = 150$ mm and a height of 220 mm, driven by two counter-rotating bladed disks.

*Comparison of the probability density distribution function (PDF) of particle angular velocity in experimental (circles) and numerical (squares) studies. The experimental results (circles) are shown for varying magnetic field strengths with fixed turbulence intensity. **b** weak (1.2mT, Supplementary Movie 3), **c** intermediate (1.4mT, Supplementary Movie 4), and **d** strong (1.6mT, Supplementary Movie 5), with constant rotational frequency of the magnetic field (parameter settings can be found in the Methods and Supplementary Information).*

... where $\omega_a = \mu\mathcal{V}_p H^2(\chi_{\parallel} - \chi_{\perp})/\xi_r$ is the normalized magnetic field intensity which is also called anisotropic frequency (Tierno et al., 2009; Címurs et al., 2019), with $\omega_a > 0$ ($\omega_a < 0$) for a prolate (oblate) particle. ~~The perpendicular rotational component, $\omega_{p,\perp}$, can be obtained by $\omega_{p,\perp} = \mathbf{n} \times \dot{\mathbf{n}}$ and represents the particle tumbling rate. ω_f is half of the vorticity of the background flow field, i.e., $\omega_f = \frac{1}{2}[\nabla \times \mathbf{u}_f]_O$ with the subscript O implying evaluation at the center of mass of the sphere, which is obtained through performing Direct Numerical Simulations (DNS) using a pseudo-spectral method and details can be found later.~~

... where for the overdamped particle, we have $\frac{d(\xi_r \omega_{p,\parallel})}{dt} = \mathbf{0}$. ω_f is half of the vorticity of the background flow field, which is obtained through performing Direct Numerical Simulations (DNS) using a pseudo-spectral method and details can be found later.

In the Method section, we remove the redundant part: ... ~~In a quiescent fluid, the critical frequency at which the particle transitions from synchronous rotation to oscillatory motion is given by: $\omega_{cr} = \frac{\omega_a}{2}$. If $\omega_H \leq \omega_{cr}$, the particle follows the rotating magnetic field synchronously with a constant phase lag β . Beyond this threshold, the particle exhibits back-and-forth oscillations.~~

Comment 11: *I believe there is a mistake in the magnetic particle density line 489. 0.2 kg/m^3 is a density 6 times lighter than air, please correct that. Why not choose neutrally buoyant particles instead of very light ones compared to water?*

Response: Indeed it should be corrected to $208 \pm 14 \text{ kg/m}^3$. We have now corrected this in the revised manuscript.

The reasons for using light particles has been discussed in the reply to Comment 2 in this rebuttal. For convenience, here we provide a brief explanation. Light particles are preferentially concentrated in the vortex filament, which offers an opportunity to control the turbulent flows from its smallest scales through external forcing. So it is essential to understand the interaction between magnetically actuated particles and the vorticity structures of turbulence. To this end, magnetic particles are designed to be light and of a size comparable to the Kolmogorov scale, allowing them to preferentially enter and interact with vortex filaments.

The modifications in the main paper are listed below:

(2) *Density:* The magnetic particles have a mean density of $\rho_{particle} = 208 \pm 14 \text{ kg/m}^3$.

Comment 12: *The bibliography is extensive (73 references). However, several experimental studies, conducted in different research groups, have investigated 3D turbulence generated by magnetic particles (e.g., E. Falcon et al. PRL 2017, A. Cazaubiel et al. Phys. Rev. Fluids 2021, J.B. Gorce et al. Phys. Rev. Lett. 2024), or regimes reminiscent of 2D turbulence (e.g., N. Francois et al. Phys. Rev. X 2014, G. Kokot et al. PNAS 2017, M. Bourgoin et al. Phys. Rev. X 2020). Including some of these works in the bibliography would strengthen the contextual background.*

Response: Following the reviewer’s suggestion, we have updated our bibliography to include the mentioned references.

The modifications in the main paper are listed below:

Previous studies have extensively explored magnetic particles in quiescent fluids under rotating magnetic fields, revealing rich phenomena such as self-assembly into chain-like structures (Martin, 2009; Shanko et al., 2019), synchronization-selected structures (Yan et al., 2012), and more complex arrays (Piet et al., 2013), enhanced mixing (Martin, 2009; Wittbracht et al., 2012; Shanko et al., 2019), turbulence design (Falcon et al., 2017; Cazaubiel et al., 2021; Gorce and Falcon, 2024) and phase locking, where particles follow the rotating magnetic field with a constant phase lag with respect to the magnetic field when the rotating magnetic angular velocity ω_H is below a threshold ω_{cr} (Cīmurs et al., 2019). Investigations have also extended to self-propelling active magnetic particles (e.g., magnetotactic bacteria) (Cēbers and Ozols, 2006; Ērglis et al., 2007), magnetic Janus particles (Yan et al., 2012; Sinn et al., 2011), particles with different aspect ratios (Cīmurs and Cēbers, 2013; Morozov et al., 2017), colloidal spinners and active swimmers (Francois et al., 2014; Kokot et al., 2017; Bourgoin et al., 2020).

Comment 13: *Is the magnetic coating uniform? There is an inconsistency on this point in the Methods section (lines 476 and lines 490-491)? What determines the direction of the magnetic moment of each particle?*

Response: We apologize for the confusion.

For convenience, we list the contents of lines 476 and lines 490-491 in the Methods section: - The information on lines 475-476: “*The magnetic paint is then applied uniformly from the top of the substrate to the Styrofoam*” - The information on lines 490-491: “*(3) Magnetic Properties: Due to the nonuniform distribution of the magnetic paint on the particle surface, the magnetic properties vary between particles.*”

What we intended to convey is the following: in the fabrication process, the magnetic paint is applied uniformly across the substrate, such that each particle is coated over its full surface. However, at the single particle level, the paint layer is not perfectly homogeneous, and small variations in thickness or distribution remain. As a result, the effective magnetic properties differ slightly between particles, and in particular the magnetic moment of a given particle is determined by the distribution of magnetic material on its surface.

Consequently, while the direction of the applied coating is uniform in a global sense (all particles are coated in the same manner), the precise direction of the magnetic moment of each individual particle is set by the asymmetries in its coating. This explains why the coating procedure is described as “uniform” in the fabrication step, yet “nonuniform” in terms of the resulting magnetic properties of the particles. We have revised the Methods section to make this distinction clearer and avoid confusion.

For clarity, here we also provide a description of the particle fabrication method (more details can be found in our experimental work (Wu et al., 2025)). The fabrication procedure is that polystyrene particles are firstly secured on a flat plastic board using adhesive tape. Magnetic paint is manually sprayed from the top left and top right at a at a 45° angle angle, coating the particles with a thin layer of magnetic material that can be magnetized in the presence of a magnetic field but exhibits no residual magnetism in free space. The non-uniform distribution of the magnetic coating on the particle surface, combined with slight deviations from a perfect spherical shape, results in magnetic anisotropy. Although it is difficult to quantitatively determine how the coating and shape jointly define the direction of the magnetic moment for individual particles, their anisotropic magnetic properties can be characterized statistically, details are reported in (Wu et al., 2025).

The modifications in the main paper are listed below:

The magnetic paint is then applied ~~uniformly~~ from the top of the substrate to the Styrofoam spheres through a spraying process, ensuring that all particles are coated in a consistent manner; however, at the single-particle level the coating is not perfectly homogeneous, and small variations in thickness or distribution lead to slight differences in magnetic properties between particles.

Minor comments: *In line 489 and the Supplemental Information, the accuracy of the magnetic particle diameter is in micrometers. Is it significant?*

Movies 6, 7, and 8 are long (5 min each) and could be sped up.

Response: The particle diameter is given in micrometers because the particles were selected using two sieves with closely spaced mesh sizes, both specified in micrometers. The particles retained between these two sieves naturally fall within a narrow micrometer-sized range. Therefore, the micrometer scale reflects the resolution of our selection method, rather than implying a high measurement accuracy.

In addition, we have sped up Movies 6, 7, and 8; each is now reduced to about 1 minute in duration.

Closing remarks

The authors thank the reviewer once more for the many detailed comments on the paper, which helped to improve the manuscript. We have implemented all the suggestions in the revised paper. We hope that with these changes and additions, the paper can now be given a full recommendation for publication in Nature Communications.

References

- Arneodo, A., Baudet, C., Belin, F., Benzi, R., Castaing, B., Chabaud, B., Chavarria, R., Ciliberto, S., Camussi, R., Chillà, F., et al. (1996). Structure functions in turbulence, in various flow configurations, at Reynolds number between 30 and 5000, using extended self-similarity. *Europhysics Letters*, 34(6):411.
- Benzi, R., Suter, A., and Vulpiani, A. (1981). The mechanism of stochastic resonance. *Journal of Physics A: Mathematical and General*, 14(11):L453.
- Bourgoin, M., Kervil, R., Cottin-Bizonne, C., Raynal, F., Volk, R., and Ybert, C. (2020). Kolmogorovian active turbulence of a sparse assembly of interacting marangoni surfers. *Physical Review X*, 10(2):021065.
- Buzzicotti, M., Biferale, L., and Toschi, F. (2020). Statistical properties of turbulence in the presence of a smart small-scale control. *Physical Review Letters*, 124(8):084504.
- Calzavarini, E., Kerscher, M., Lohse, D., and Toschi, F. (2008). Dimensionality and morphology of particle and bubble clusters in turbulent flow. *Journal of Fluid Mechanics*, 607:13–24.
- Cazaubiel, A., Gorce, J.-B., Bacri, J.-C., Berhanu, M., Laroche, C., and Falcon, E. (2021). Three-dimensional turbulence generated homogeneously by magnetic particles. *Physical Review Fluids*, 6(11):L112601.
- Cēbers, A. and Ozols, M. (2006). Dynamics of an active magnetic particle in a rotating magnetic field. *Physical Review E*, 73(2):021505.

- Chen, P., Weady, S., Atis, S., Matsuzawa, T., Shelley, M. J., and Irvine, W. T. (2025). Self-propulsion, flocking and chiral active phases from particles spinning at intermediate reynolds numbers. Nature Physics, 21(1):146–154.
- Cīmurs, J., Brasovs, A., and Ērglis, K. (2019). Stability analysis of a paramagnetic spheroid in a precessing field. Journal of Magnetism and Magnetic Materials, 491:165630.
- Cīmurs, J. and Cēbers, A. (2013). Dynamics of anisotropic superparamagnetic particles in a precessing magnetic field. Physical Review E, 87(6):062318.
- Croquette, V. and Poitou, C. (1981). Cascade of period doubling bifurcations and large stochasticity in the motions of a compass. Journal de Physique Lettres, 42(24):537–539.
- da Silva, C. B., Dos Reis, R. J., and Pereira, J. C. (2011). The intense vorticity structures near the turbulent/non-turbulent interface in a jet. Journal of Fluid Mechanics, 685:165–190.
- Ērglis, K., Wen, Q., Ose, V., Zeltins, A., Sharipo, A., Janmey, P. A., and Cēbers, A. (2007). Dynamics of magnetotactic bacteria in a rotating magnetic field. Biophysical Journal, 93(4):1402–1412.
- Falcon, E., Bacri, J.-C., and Laroche, C. (2017). Dissipated power within a turbulent flow forced homogeneously by magnetic particles. Physical Review Fluids, 2(10):102601.
- Francois, N., Xia, H., Punzmann, H., Ramsden, S., and Shats, M. (2014). Three-dimensional fluid motion in faraday waves: creation of vorticity and generation of two-dimensional turbulence. Physical Review X, 4(2):021021.
- Freitas, A., de Wit, X. M., Wang, Z., Biferale, L., and Toschi, F. (2025). Statistical properties of turbulence under a smart lagrangian forcing. arXiv preprint arXiv:2508.06660.
- Frisch, U. (1995). Turbulence: the legacy of AN Kolmogorov. Cambridge university press.
- Fruchart, M., Scheibner, C., and Vitelli, V. (2023). Odd viscosity and odd elasticity. Annual Review of Condensed Matter Physics, 14(1):471–510.
- Gammaitoni, L., Hänggi, P., Jung, P., and Marchesoni, F. (1998). Stochastic resonance. Reviews of Modern Physics, 70(1):223.
- Ganapathisubramani, B., Lakshminarasimhan, K., and Clemens, N. (2008). Investigation of three-dimensional structure of fine scales in a turbulent jet by using cinematographic stereoscopic particle image velocimetry. Journal of Fluid Mechanics, 598:141–175.
- Ghira, A., Elsinga, G., and Da Silva, C. (2022). Characteristics of the intense vorticity structures in isotropic turbulence at high reynolds numbers. Physical Review Fluids, 7(10):104605.

- Gorce, J.-B. and Falcon, E. (2024). Freely decaying saffman turbulence experimentally generated by magnetic stirrers. Physical Review Letters, 132(26):264001.
- Jiang, S., Li, B., Zhao, J., Wu, D., Zhang, Y., Zhao, Z., Zhang, Y., Yu, H., Shao, K., Zhang, C., et al. (2023). Magnetic janus origami robot for cross-scale droplet omni-manipulation. Nature Communications, 14(1):5455.
- Kaiser, A., Snezhko, A., and Aranson, I. S. (2017). Flocking ferromagnetic colloids. Science advances, 3(2):e1601469.
- Kang, S.-J., Tanahashi, M., and Miyauchi, T. (2007). Dynamics of fine scale eddy clusters in turbulent channel flows. Journal of Turbulence, 8:N52.
- Kokot, G., Das, S., Winkler, R. G., Gompper, G., Aranson, I. S., and Snezhko, A. (2017). Active turbulence in a gas of self-assembled spinners. Proceedings of the National Academy of Sciences, 114(49):12870–12875.
- La Porta, A., Voth, G. A., Crawford, A. M., Alexander, J., and Bodenschatz, E. (2001). Fluid particle accelerations in fully developed turbulence. Nature, 409(6823):1017–1019.
- Labbé, R., Pinton, J.-F., and Fauve, S. (1996). Study of the von kármán flow between coaxial corotating disks. Phys. Fluids, 8(4):914–922.
- Martin, J. E. (2009). Theory of strong intrinsic mixing of particle suspensions in vortex magnetic fields. Physical Review E, 79(1):011503.
- Mathai, V., Lohse, D., and Sun, C. (2020). Bubbly and buoyant particle-laden turbulent flows. Annual Review of Condensed Matter Physics, 11:529–559.
- Mordant, N. (1997). Characterization of turbulence in a closed flow. J. Phys. II France, 7:1729–1742.
- Mordant, N., Lévêque, E., and Pinton, J.-F. (2004). Experimental and numerical study of the lagrangian dynamics of high reynolds turbulence. New Journal of Physics, 6(1):116.
- Mordant, N., Metz, P., Michel, O., and Pinton, J.-F. (2001). Measurement of lagrangian velocity in fully developed turbulence. Physical Review Letters, 87(21):214501.
- Morozov, K. I., Mirzae, Y., Kenneth, O., and Leshansky, A. M. (2017). Dynamics of arbitrary shaped propellers driven by a rotating magnetic field. Physical Review Fluids, 2(4):044202.
- Piet, D., Straube, A., Snezhko, A., and Aranson, I. (2013). Viscosity control of the dynamic self-assembly in ferromagnetic suspensions. Physical Review Letters, 110(19):198001.
- Poyé, A., Désangles, V., Jiménez, X., Martin, M., and Proto, Y. (2018). Bipolar motor: rotation, parametric instabilities and chaos. Physica Scripta, 94(1):015002.

- Scholz, C., Engel, M., and Pöschel, T. (2018). Rotating robots move collectively and self-organize. Nature Communications, 9(1):931.
- Shanko, E.-S., van de Burgt, Y., Anderson, P. D., and den Toonder, J. M. (2019). Microfluidic magnetic mixing at low reynolds numbers and in stagnant fluids. Micromachines, 10(11):731.
- Sinn, I., Kinnunen, P., Pei, S. N., Clarke, R., McNaughton, B. H., and Kopelman, R. (2011). Magnetically uniform and tunable janus particles. Applied Physics Letters, 98(2):024101.
- Stroock, A. D., Dertinger, S. K., Ajdari, A., Mezic, I., Stone, H. A., and Whitesides, G. M. (2002). Chaotic mixer for microchannels. Science, 295(5555):647–651.
- Tanahashi, M., Iwase, S., and Miyauchi, T. (2001). Appearance and alignment with strain rate of coherent fine scale eddies in turbulent mixing layer. Journal of Turbulence, 2(1):006.
- Tierno, P., Claret, J., Sagués, F., and Cēbers, A. (2009). Overdamped dynamics of paramagnetic ellipsoids in a precessing magnetic field. Physical Review E, 79(2):021501.
- Toms, B. A. (1949). Some observations on the flow of linear polymersolutions through straight tubes at large reynolds numbers. In Proc. 1st Intl Congr. Rheol., volume 2, pages 135–141.
- van den Berg, T. H., van Gils, D. P., Lathrop, D. P., and Lohse, D. (2007). Bubbly turbulent drag reduction is a boundary layer effect. Physical review letters, 98(8):084501.
- Van Gils, D. P., Guzman, D. N., Sun, C., and Lohse, D. (2013). The importance of bubble deformability for strong drag reduction in bubbly turbulent taylor–couette flow. Journal of fluid mechanics, 722:317–347.
- Volk, R., Calzavarini, E., Leveque, E., and Pinton, J.-F. (2011). Dynamics of inertial particles in a turbulent von kármán flow. Journal of Fluid Mechanics, 668:223–235.
- Volk, R., Calzavarini, E., Verhille, G., Lohse, D., Mordant, N., Pinton, J.-F., and Toschi, F. (2008). Acceleration of heavy and light particles in turbulence: comparison between experiments and direct numerical simulations. Physica D: Nonlinear Phenomena, 237(14-17):2084–2089.
- Voth, G. A., La Porta, A., Crawford, A. M., Alexander, J., and Bodenschatz, E. (2002). Measurement of particle accelerations in fully developed turbulence. Journal of Fluid Mechanics, 469:121–160.
- Wang, Z., de Wit, X. M., and Toschi, F. (2024). Localization–delocalization transition for light particles in turbulence. Proceedings of the National Academy of Sciences, 121(38):e2405459121.
- Wittbracht, F., Weddemann, A., Eickenberg, B., Zahn, M., and Hütten, A. (2012). Enhanced fluid mixing and separation of magnetic bead agglomerates based on dipolar interaction in rotating magnetic fields. Applied Physics Letters, 100(12).

- Wu, C., Kunnen, R. P. J., Wang, Z., de Wit, X. M., Toschi, F., and Clercx, H. J. H. (2025). Tracking the rotation of light magnetic particles in turbulence. arXiv preprint arXiv:2506.21769.
- Yan, J., Bloom, M., Bae, S. C., Luijten, E., and Granick, S. (2012). Linking synchronization to self-assembly using magnetic janus colloids. *Nature*, 491(7425):578–581.
- Yang, L. and Zhang, L. (2021). Motion control in magnetic microrobotics: From individual and multiple robots to swarms. *Annual Review of Control, Robotics, and Autonomous Systems*, 4(1):509–534.
- Zocchi, G., Tabeling, P., Maurer, J., and Willaime, H. (1994). Measurement of the scaling of the dissipation at high reynolds numbers. *Physical Review E*, 50(5):3693.

Reply to Reviewer 2

Synopsis

The author considered most of my suggestions, including adding experimental data, changing the title, and focusing the scope of the study on stochastic resonance, which is the main result, rather than the phase-locked or back-and-forth regimes already explored. Moreover, the authors have added more experimental details, but some comments are still lacking.

The revised manuscript has been improved and is now ready for publication in Nature Communications, provided the authors address the points outlined below.

Reply

We thank the Reviewer for their positive assessment of our work. Below we address the feedback of the Reviewer 2 point-by-point. The resulting changes to the text are marked in blue in the manuscript.

Comment i

In contrast to the response given to my Comment 2, the new experimental points performed do not appear in Fig. 3(m) as in Fig. R1 for Reviewer 2. I think it is just an omission.

Reply

We thank the Reviewer for reminding us. We have now added the new experimental data in Fig. 3(m) in the revised manuscript.

The updated Fig.3 in the main text is now as follows:

Comment ii

The authors still claim that this concept has strong analogies with magnetometry techniques used in biomedical applications, but do not give any references or explain how they can scale up this technique.

Reply

We now clarify better as follows.

Our previous argument in the reply to Comment 4 from Reviewer 2 said that: “ *the magnetic signature emitted by the particles themselves (e.g., via magnetic dipole field measurements) can be detected by magnetic sensors placed outside the flow. In such a setup, the rotation dynamics of the particles can be reconstructed from the modulated magnetic field they emit, making optical access unnecessary*”. We think this concept is similar to the Magnetic Particle Spectroscopy (MPS) techniques used in biomedical applications.

As discussed in the recent review paper [1], MPS relies on the nonlinear dynamic magnetic response of superparamagnetic iron oxide nanoparticles (SPIONs) subjected to an oscillating magnetic field. The magnetic moments of these particles attempt to align with the applied magnetic field, similar to the conditions experienced by the magnetic particles in our setup. Due to the nonlinear nature of this process, the magnetic response of the particles contains higher harmonics, whose amplitude and phase are highly sensitive to the particle dynamic behavior and their surrounding environment, such as the fluid viscosity. So, by remotely measuring and analyzing these magnetic responses, it is possible to extract information about the particle dynamics or the properties of the surrounding environment. By analogy, in our system, magnetic particles are subjected to a dynamically varying magnetic field and rotate in response to turbulent vorticity, which can be thought of as a dynamic and complex *environment* acting on the magnetic particles (probes). We propose that, similar to MPS, their magnetic response could in principle be detected. By pre-calibrating the relationship between the particle angular dynamics and their magnetic response, it would be possible to infer the rotational behavior of the particles in the flow without relying on optical tracking. We understand the confusion and acknowledge that this is currently only a conceptual analogy and an outlook for potential extension. Applying this concept to practical biomedical detection would require further detailed investigation beyond the scope of the current work.

We have now added more explanations on this concept in the revised main text.

The modification in the main text (conclusion):

In the present work, we validate this concept using optical measurements of particle rotation. However, in principle, once the angular response is well-characterized, the same protocol could be applied even in optically inaccessible environments by measuring the magnetic fields emitted by the spinning magnetic particle directly. This concept shares similar principles with Magnetic Particle Spectroscopy (MPS), a technique that uses the dynamic magnetic responses of magnetic particles to remotely sense properties of their surrounding environment [1]. In such scenarios, the activated magnetic particles would function as remotely readable vorticity probes.

Comment iii

The Stokes number asked in my Comment 5 is still not given.

Reply

In the experiments, for the turbulent flow with a Taylor Reynolds number of $Re_\lambda = 398$, the Stokes number is $St = d_p^2/(12\beta_p\nu\tau_\eta) = 4.36$. In the stronger turbulence case with $Re_\lambda = 447$, the Stokes number increases to $St = 6.08$.

We have now added these details in the revised Supplementary Information.

The modification in the Supplementary Information:

For the weak-turbulence case, the impellers rotate at 0.83 Hz. The corresponding mean energy dissipation rate is $\varepsilon = 0.037 \pm 0.002 \text{ m}^2/\text{s}^3$, which yields a Taylor-scale Reynolds number of $Re_\lambda = 398$, a Kolmogorov length scale of $\eta = 0.072 \text{ mm}$, and a Kolmogorov time scale of $\tau_\eta = 5.20 \text{ ms}$. The corresponding particle Stokes number is \$St = d_p^2/(12\beta_p\nu\tau_\eta) = 4.36\$ For the strong-turbulence case, the impellers rotate at 1.21 Hz. The corresponding mean energy dissipation rate is $\varepsilon = 0.075 \pm 0.001 \text{ m}^2/\text{s}^3$, which yields a Taylor-scale Reynolds number of $Re_\lambda = 447$, a Kolmogorov length scale of $\eta = 0.061 \text{ mm}$, and a Kolmogorov time scale of $\tau_\eta = 3.66 \text{ ms}$. The corresponding particle Stokes number is \$St = 6.08\$.

Comment iv

The sentence, answering my Comment 11, “to this end, magnetic particles are designed to be light and of a size comparable to the Kolmogorov scale, allowing them to preferentially enter and interact with vortex filaments” is an important statement that should be clearly emphasized in the manuscript.

Reply

We thank the Reviewer for reminding us. We have now added this information in the revised manuscript.

The modification in the main text (Introduction section):

Our experimental platform (Fig. 1a and Fig. 1e) involves a dilute collection of approximately spherical light magnetic particles (produced by Styrofoam core coated with magnetic paint, Fig. 1b, with a mean density of 20% with respect to water and with a mean diameter of 0.7 mm, see Methods for details) which are immersed in a Von Kármán-type turbulent flow (producing homogeneous and isotropic turbulence (HIT) in the center region of the cell [2, 3, 4, 5]) in an octagonal water-filled chamber. Magnetic particles are designed to be light and of a size comparable to the Kolmogorov scale, allowing them to preferentially enter and interact with vortex filaments.

Comment v

The Comment 14 is still not answered: Is it in Nature Communications' policies to publish the dedicated instrumentation and methodology of this manuscript in another journal – see new Ref. [60] (experimental) and Ref. [54] (numerics) of the same authors' group? Furthermore, the reliance on arXiv publications posted after the first round of review raises concerns about the transparency of the revision process.

Reply

We understand the question from the Reviewer, but we want to point out that the scopes of the works Ref. [60] and Ref. [54] are markedly distinct from the current paper. The current work focuses on the physics of (magnetically) rotating particles in turbulence and specifically the stochastic resonance mechanism that arises from their dynamics.

The paper Ref. [60] only describes and validates the experimental platform. This platform is used in the current work, but will also be used for a variety of future works. We believe that the experimental and simulation details provided within the current manuscript here are sufficient to fully substantiate the physics we describe and ensure the reproducibility of our results. The detailed validation and additional considerations of the specific experimental design are not relevant to the core physics of the current paper and we hence deemed it more appropriate to address this in a separate methodological paper. We sincerely apologize for the lack of clarity caused by the fact that the experimental paper appeared on arXiv only after the current paper was submitted for review, which was the consequence of some practical constraints on our side.

Ref. [54] has a different scope and looks at manipulating turbulence with small particles that are oscillating (rather than rotating). While it does fit in the same broader line of research about probing and manipulating turbulence with small particles, it is fundamentally unrelated to the stochastic resonance mechanism discussed in the current paper.

Minor

In the Theoretical model Section, the sentence “and details can be found later” should be removed.

Reply

We thank the Reviewer for reminding us. We have now removed this sentence.

The modification in the main text (Theoretical model Section):

... where for the overdamped particle, we have $\frac{d(\xi_r \omega_{p,\parallel})}{dt} = \mathbf{0}$. ω_f is half of the vorticity of the background flow field, which is obtained through performing Direct Numerical Simulations (DNS) using a pseudo-spectral method ~~and details can be found later~~. The validity of the overdamped model is further discussed in the Supplementary Information.

We thank the Reviewer once more for their valuable comments.

References

- [1] Kai Wu, Diqing Su, Renata Saha, Jinming Liu, Vinit Kumar Chugh, and Jian-Ping Wang. “Magnetic particle spectroscopy: a short review of applications using magnetic nanoparticles”. In: *ACS Applied Nano Materials* 3.6 (2020), pp. 4972–4989.
- [2] Nicolas Mordant, Emmanuel Lévêque, and Jean-Francois Pinton. “Experimental and numerical study of the Lagrangian dynamics of high Reynolds turbulence”. In: *New Journal of Physics* 6.1 (2004), p. 116.
- [3] Romain Volk, Enrico Calzavarini, Gautier Verhille, Detlef Lohse, Nicolas Mordant, J-F Pinton, and Federico Toschi. “Acceleration of heavy and light particles in turbulence: comparison between experiments and direct numerical simulations”. In: *Physica D: Non-linear Phenomena* 237.14-17 (2008), pp. 2084–2089.
- [4] Greg A Voth, Arthur La Porta, Alice M Crawford, Jim Alexander, and Eberhard Bodenschatz. “Measurement of particle accelerations in fully developed turbulence”. In: *Journal of Fluid Mechanics* 469 (2002), pp. 121–160.
- [5] Romain Volk, Enrico Calzavarini, Emmanuel Leveque, and J-F Pinton. “Dynamics of inertial particles in a turbulent von Kármán flow”. In: *Journal of Fluid Mechanics* 668 (2011), pp. 223–235.